# RAVEN: END-TO-END EQUIVARIANT ROBOT LEARNING WITH RGB CAMERAS

**David M. Klee**[*, 1], **Boce Hu**[*, 1], **Andrew Cole**[1], **Heng Tian**[1], **Dian Wang**[2],
**Robert Platt**[†, 1], **Robin Walters**[†, 1]

[1]Northeastern University, [2]Stanford University, [*] Equal Contribution, [†]Equal Advising

{klee.d, hu.boce, r.platt, r.walters}@northeastern.edu

## ABSTRACT

Recent work has shown that equivariant policy networks can achieve strong performance on robot manipulation tasks with limited human demonstrations. However, existing equivariant methods typically require structured inputs, such as 3D point clouds or top-down camera views, which prevents their use in low-cost setups or dynamic environments. In this work, we propose the first SE(3)-equivariant policy learning framework that operates with only RGB image observations. The key insight is to treat image-based data as collections of rays that, unlike 2D pixels, transform under 3D roto-translations. Extensive experiments in both simulation with diverse robot configurations and real-world settings demonstrate that our method consistently surpasses strong baselines in both performance and efficiency. Our project page is available at https://dmklee.github.io/raven.

## 1 INTRODUCTION

Learning effective visuomotor policies for robotic manipulation remains a fundamental challenge, particularly in settings that demand strong generalization from limited data. Recent advances in diffusion-based policy learning (Chi et al., 2023; Wang et al., 2024; Ze et al., 2024) and large-scale behavior cloning pipelines (Barreiros et al., 2025; Black et al., 2024), have demonstrated remarkable performance across diverse scenes and viewpoints. However, these methods typically require large amounts of training data to be robust to sensor or scene variations, which limits their practicality in settings where data collection is expensive or the environment is unstructured.

A promising direction to improve generalization and sample efficiency is through equivariance, which explicitly encodes symmetries into networks (Cohen & Welling, 2016). By ensuring that policies respond consistently under spatial transformations of the input, equivariant models generalize more effectively across variations in object pose and scene layout. However, existing equivariant methods impose strict requirements on input modalities, typically assuming access to structured geometric data such as point clouds or bird's-eye view RGB-D images with known alignment (Yang et al., 2024; Wang et al., 2024). Unfortunately, these assumptions are violated in many popular robot learning setups, where raw image observations from arbitrarily placed cameras (as in UMI (Chi et al., 2024), ALOHA (Zhao et al., 2023), Mobile-ALOHA (Fu et al., 2024)) are the norm.

Our method, **RA**y-based equi**V**ariant **EN**coder or RAVEN, is the first end-to-end SE(3) equivariant policy learning method that works with image-based observations. Our key insight is to express images in terms of rays projected into 3D space using camera parameters. In particular, each small patch of pixels in the image is encoded as a feature vector with a corresponding ray originating at the camera aperture and passing through the patch. Unlike raw pixels that exist only in the camera frame and therefore only support 2D transformations, rays are naturally defined in 3D and can be transformed by SE(3) rotations and translations. This gives us a representation of the input images that is compatible with SE(3)-equivariant neural network layers and therefore enables us to create an end-to-end SE(3)-equivariant model using only RGB input while remaining agnostic to the number and placement of cameras in the robot learning setup.

RAVEN utilizes a novel ray-based observation encoder that transforms multi-view images into SE(3)-equivariant geometric tokens. To decode these tokens into actions, we introduce a lightweight and

flexible policy head that supports both world-frame (absolute) and end-effector-frame (relative) action representations, allowing compatibility with a wide range of robotic systems and control interfaces. Unlike prior equivariant methods that often suffer from slow training due to heavy architectures, our method achieves $SE(3)$-equivariance with a training speed that is even faster than a baseline Diffusion Policy, but with significantly improved performance.

This paper makes the following contributions:

- We introduce an encoder that expresses an image as a set of $SE(3)$-equivariant geometric tokens. Tokens from different, arbitrarily placed cameras can be seamlessly combined to produce a single, consistent representation of the world.

- Based on this encoder, we propose RAVEN, the first end-to-end $SE(3)$-equivariant robotic policy learning framework that can operate directly on RGB image inputs.

- We characterize our method both in simulation and on a physical robot. It outperforms the strongest baseline by 12% over 12 MimicGen tasks, 17% over 6 DexMimicGen tasks, and 35% over 4 real-world tasks. Finally, while prior equivariant models are often slow to train, our method trains approximately $1.6\times$ faster than the previous equivariant diffusion method.

## 2 RELATED WORK

**Equivariant Networks for Image Processing**  Equivariant models for 2D image processing (Cohen & Welling, 2016; Dieleman et al., 2016; Worrall et al., 2017; Chidester et al., 2018; Weiler & Cesa, 2019; Maron et al., 2020; He et al., 2021; Rahman & Yeh, 2023; Li et al., 2024; 2025) achieve equivariance to planar transformation groups, including $SO(2)$, $SE(2)$, and more general affine groups, which improves generalization and sample efficiency in vision tasks involving spatial symmetries. Extending beyond such planar symmetry, recent efforts (Esteves et al., 2019; Park et al., 2022; Klee et al., 2023a;b; Howell et al., 2023; Lee & Cho, 2024) seek to learn $SO(3)$- or $SE(3)$-equivariant representations directly from 2D images for 3D vision tasks, such as 6D pose estimation and object understanding. These methods attempt to bridge the gap between 2D observation and 3D geometric reasoning by encoding 3D symmetries into the learned features. More recently, ray-based formulations have emerged as a promising direction for incorporating 3D geometry into image-based equivariant models. By interpreting each pixel as a 3D ray originating from the camera, these approaches anchor features along these rays in 3D space while maintaining equivariance (Xu et al., 2023; Brehmer et al., 2024; Xu et al., 2024). Our work builds on GTA (Miyato et al., 2023), which introduces an efficient attention mechanism with camera pose and pixel coordinate geometric representations. In contrast, we represent image patches as ray-based $SE(3)$ representations, which better encode the spread of rays in the scene and allow processing with non-image data.

**Equivariant Robotic Manipulation**  Symmetry has been widely explored in robotic manipulation to improve generalization and sample efficiency (Huang et al., 2022; 2024; Wang et al., 2022; Nguyen et al., 2023; Zhu et al., 2022; Hu et al., 2024a; Jia et al., 2022; Ryu et al., 2022; 2024; Gao et al., 2024; Mittal et al., 2024; Eisner et al., 2024). Recent works have transitioned from open-loop to closed-loop control by adopting generative policies such as diffusion models (Chi et al., 2023) and flow matching approaches (Hu et al., 2024b; Chisari et al., 2024). Within this framework, incorporating symmetry further improves performance. Wang et al. (2024) introduces an $SO(2)$-equivariant diffusion policy, but assumes fixed camera placement. EquiBot (Yang et al., 2024) achieves $SIM(3)$-equivariance using object-centric point clouds, but relies on accurate object segmentation, which limits generalization to diverse sensor configurations and open-world setups. More recently, Hu et al. (2025) proposes an $SO(3)$-equivariant policy using only RGB inputs from a single wrist-mounted camera. Wang et al. (2025) further highlights the benefits of pretrained vision encoders and relative action spaces in diffusion-based policies. In contrast, RAVEN is the first, as far as we know, $SE(3)$-equivariant manipulation framework that operates directly on raw RGB images without relying on object segmentation or assumptions about the number and placement of cameras. Additionally, our model achieves equivariance through canonicalization and is trained via flow-matching, so our model is less computationally heavy than existing equivariant methods.

## 3    BACKGROUND

**Equivariance and Canonicalization**    A function $f$ is equivariant to a symmetry group $G$ if $\rho'_g f(x) = f(\rho_g x)$ for all $g \in G, x \in X$ where $\rho$ and $\rho'$ denote group actions on the input and output spaces, respectively. Equivariance is desirable in deep learning because it allows models to generalize across transformations of the data. Equivariant networks can be constructed through group convolution (Cohen & Welling, 2016; Esteves et al., 2019) or by imposing constraints on the weights of each layer, as shown by (Geiger & Smidt, 2022; Cesa et al., 2022; Brehmer et al., 2024).

An alternative approach is to enforce equivariance with canonicalization: first, transform each input into a standardized, or canonical, reference frame that is consistent across samples, and then map the output back to the original frame. Canonicalization avoids the computational overhead of equivariant convolutions, but assumes access to a useful canonicalizer. In practice, such canonicalization functions can be defined analytically (e.g., aligning to principal axes), estimated (e.g., via PCA), or learned from data (Esteves et al., 2017; Du et al., 2022; Kaba et al., 2023). Formally, let $G$ act on the input space $X$ via $\rho : G \times X \to X$, and let $\mathcal{C} : X \to G$ be a canonicalization function satisfying the right-compensation property:

$$\mathcal{C}(\rho_g x) = \mathcal{C}(x)g^{-1}, \qquad \forall\, g \in G,\ x \in X. \tag{1}$$

We denote the canonicalized input by $\rho_{\mathcal{C}(x)}x$. Using (1), we have: $\rho_{\mathcal{C}(\rho_g x)}(\rho_g x) = \rho_{\mathcal{C}(x)}x$, so $x \mapsto f(\rho_{\mathcal{C}(x)}x)$ is $G$-invariant. To recover equivariance, we decanonicalize in the output space with action $\rho'$: $F(x) = \rho'_{\mathcal{C}(x)^{-1}} f(\rho_{\mathcal{C}(x)}x)$ which implies $F(\rho_g x) = \rho'_g F(x)$.

**Images and Rays**    An image is typically represented as a discretized 2D signal, that is, a mapping from pixel coordinates to RGB values, $I : \mathbb{Z}^2 \mapsto \mathbb{R}^3$. Alternatively, an image can be thought of as a collection of rays moving from the camera's optical center out into the world. This ray-based representation better conveys 3D spatial information of the world. The ray associated with a 2D pixel coordinate is determined by the camera intrinsics $K \in \mathbb{R}^{3\times 3}$ and extrinsics $M = [R|t] \in \mathrm{SE}(3)$, where $R \in \mathrm{SO}(3)$ and $t \in \mathbb{R}^3$ denote the camera orientation and position in the world frame. Given a pixel coordinate $u \in \mathbb{Z}^2$ expressed in homogeneous coordinates, we can compute the corresponding ray $r$ as:

$$r = (t, RK^{-1}u) \tag{2}$$

where $r = (r_o, r_d)$ is a tuple describing the origin $r_o \in \mathbb{R}^3$ and direction $r_d \in S^2$. This follows from the standard camera projection equation, which maps a 3D point $x \in \mathbb{R}^3$ to a pixel coordinate $u \in \mathbb{Z}^2$: $u = K[R^T| - R^T t]x$. This analysis is based on a pinhole camera model; for other models, such as fisheye cameras, we refer the reader to Grossberg & Nayar (2001); Kannala & Brandt (2006).

**Geometric Transform Attention**    Geometric Transform Attention (GTA) is a recently proposed attention mechanism designed for spatial data (Miyato et al., 2023) to achieve equivariance. Specifically, each token $x_i$ is associated with a group element $g_i \in G$ that encodes its geometric context (in the simplest case, an image, the context is the 2D pixel location; but the idea extends to $\mathrm{SE}(3)$ poses of cameras, etc.). Given some representation $\rho : G \to GL_d(\mathbb{R})$ that maps group elements to linear operators, the output of geometric transform attention between the $i$-th query, $Q_i$, and the $j$-th key and value, $K_j, V_j$, is:

$$O_i = \rho(g_i)\frac{\exp\left((\rho(g_i)^\top Q_i)^\top (\rho(g_j)^{-1}K_j)\right)}{\sum_{j'=1}^{n} \exp\left((\rho(g_i)^\top Q_i)^\top (\rho(g_{j'}^{-1})K_{j'})\right)}(\rho(g_j)^{-1}V_j). \tag{3}$$

This construction ensures that attention scores are only affected by the relative transformation $\rho_{g_i}^\top \rho_{g_j}$ between tokens. Thus, if all tokens undergo the same global transformation ($g_i \mapsto hg_i$ for some $h \in G$), the internal similarity remains unchanged and the output transforms consistently by $\rho(h)$, yielding equivariance. As noted by the authors, dot-product similarity violates equivariance when $G = \mathrm{SE}(3)$, so they propose a variation on Equation 3 that uses negative Euclidean distance as a similarity metric to achieve exact $\mathrm{SE}(3)$ equivariance.

**Policy Learning with Flow Matching**    Flow-based generative modeling provides an alternative to diffusion for capturing multimodal action distributions in policy learning. Unlike DDPM (Ho et al.,

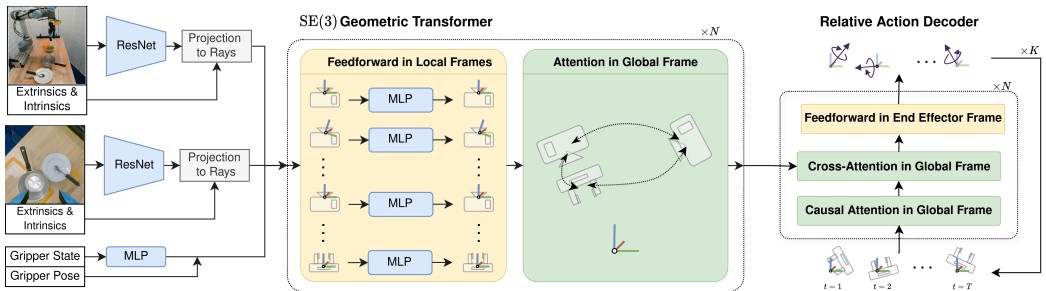

Figure 1: The RAVEN Architecture: image observations are converted to geometric tokens by preprocessing with ResNet and assigning pose based on rays. The SE(3) Geometric Transformer efficiently processes the tokens with feedforward and geometric transform attention layers. Finally, an action decoder network refines relative end-effector motions with flow matching.

2020), which predicts noise in a discrete denoising process, flow matching (Lipman et al., 2022; Liu et al., 2022) directly learns a continuous-time vector field that transports a prior distribution to the expert action distribution. Concretely, the policy is represented as an ordinary differential equation (ODE): $d\mathbf{a}(t)/dt = v_\theta(\mathbf{o}, \mathbf{a}(t), t)$ where $\mathbf{a}(0) \sim \mathcal{N}(0, I)$ and the terminal state $\mathbf{a}(1)$ follows the expert action distribution. During training, $\mathbf{a}(t)$ is approximated by interpolating between a noise sample $\mathbf{a}_0 \sim p_0$ and an expert action $\mathbf{a}_1 \sim p_1$, and the model $v_\theta$ is supervised with the flow matching objective:

$$\mathcal{L}_{\text{FM}} = \mathbb{E}_{t, \mathbf{a}_0 \sim p_0, \mathbf{a}_1 \sim p_1} \left[ \| v_\theta \big( (1 - t)\mathbf{a}_0 + t\mathbf{a}_1, t \big) - (\mathbf{a}_1 - \mathbf{a}_0) \|^2 \right] \tag{4}$$

which trains $v_\theta$ to approximate the transport field that maps the prior $p_0$ to the expert distribution $p_1$. Intuitively, $v_\theta(\mathbf{a}, t)$ tells each point how to move over time, and this direct supervision enables more efficient sampling and more stable training compared to diffusion-based policies. As shown in similar works on diffusion policy (Wang et al., 2024; Yang et al., 2024), flow matching can be made SE(3) equivariant if the velocity field is SO(3) equivariant and the observations are centered Chisari et al. (2024).

## 4 METHOD

Our method, RAVEN, is a policy learning framework for image-based observations that is equivariant to SE(3) transformations of the scene (e.g. robots, objects, and sensors). Unlike existing equivariant methods, RAVEN does not require spatial data like point clouds and can be used with any number of stationary or moving RGB cameras. In this section, we first describe a novel equivariant encoder network that digests images and proprioceptive information to produce a dense, equivariant representation of the scene. Second, we introduce an action decoder network that operates on this observation encoding to produce action trajectories. Finally, we analyze the equivariance properties of the method.

### 4.1 RAY-BASED EQUIVARIANT ENCODER

We propose an SE(3) equivariant encoder network that efficiently processes multi-image and proprioceptive observations. First, we map 2D image data to 3D ray features, which better convey spatial information about the scene. Next, we express all input data in a unified representation, called *geometric tokens*, in which the features are canonicalized by their pose in the scene. Finally, we process these tokens with GTA-style transformer layers, composed of feedforward operations in local frames and attention operations in the global frame.

**Ray-based** SE(3) **Features from Images** It is difficult to integrate image inputs into 3D equivariant methods, since the 3D roto-translation action on the data is unclear. Inspired by recent works (Brehmer et al., 2024; Xu et al., 2024), we instead consider each pixel to represent a 3D ray, according to the known camera extrinsics and intrinsics. In this way, an SE(3) transformation acts on the extrinsics (camera pose), which transforms the rays even though the image data is unchanged.

In practice, processing 3D data like rays is expensive, so we propose first pre-processing the images with a pretrained ResNet (He et al., 2016). This is useful for two reasons: (1) pretrained image models are fast and effective at extracting useful information, (2) the resulting feature map is downsampled, which reduces the burden of subsequent operations on 3D data. However, after downsampling, each grid cell in the feature map represents a local patch of pixels and a corresponding local patch of rays. While a single ray is symmetric about the camera's roll axis, a local patch of rays has a well-defined orientation. In this way, each grid cell in the feature map is associated with an element of $\mathrm{SE}(3)$, describing its origin and orientation (direction is z-axis, and image plane axes are x, y-axes). The following subsections explain how to use these features and their poses for fast and equivariant processing.

**Geometric Tokens** A geometric token is a piece of geometric information stored in a reference frame. Specifically, a geometric token is a tuple, $x = (z_x, g_x)$, composed of a feature vector, $z_x \in \mathbb{R}^d$, and a pose, $g_x \in \mathrm{SE}(3)$, such that the feature vector is canonicalized by the pose, e.g., it conveys local information. The idea is similar to positional embeddings in ViT's, where each token has features describing a local patch which may be combined with the coordinate of the patch via a positional embedding. The idea was described in its most general form by Miyato et al. (2023), where each token has features and a group element. In contrast, our geometric tokens only contain $\mathrm{SE}(3)$ group elements, providing a unified representation across different input modalities.

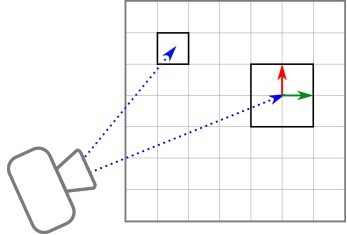

Figure 2: A pixel conveys a ray direction (left), an image patch conveys an orientation (right).

A geometric token's pose is used to convert the features into a global frame: $\rho(g_x)z_x$. The feature vector, $z_x$, is a combination of scalar, $z_x^s$, vector, $z_x^v$, and point, $z_x^p$ features so that:

$$\rho(g_x)z_x = (\rho^s(g_x)z_x^s, \rho^v(g_x)z_x^v, \rho^p(g_x)z_x^p) \tag{5}$$

where $\rho^s(g_x)$ is the identity (scalar features are invariant to geometric transforms), $\rho^v(g_x)$ applies 3D rotation and $\rho^p(g_x)$ applies 3D rotation and translation. We illustrate this idea in Figure 3. A helpful way to think of geometric tokens is through the lens of canonicalization. The features are canonicalized by default, and the pose is used to de-canonicalize them using a hardcoded $\mathrm{SE}(3)$ group action. This ability to process information in canonical or decanonical frames is key to our method's $\mathrm{SE}(3)$ equivariance and computational efficiency.

A core assumption here is that all observations can be converted to geometric tokens. For image observations, we discussed how we use a ResNet to generate a feature map where each grid cell is assigned a pose based on the camera projection. We apply a 1x1 convolution to the feature map to obtain the desired feature dimensionality (split among scalar, vector, and point features). For the proprioceptive information, e.g., end-effector pose and state (gripper opening or finger joint angles), we construct one geometric token per robot. The features are generated with a linear mapping on the gripper state, and the pose is defined as the end-effector pose. We discuss tokenization schemes for other observation modalities in Appendix C.

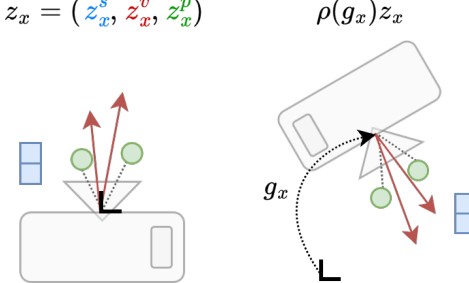

Figure 3: A geometric token encodes scalar, vector, and point features in the sensor's local frame (left), which can be transformed to the global frame with the token's pose (right).

$\mathrm{SE}(3)$ **Geometric Transformer** Once all observations are converted into geometric tokens, we process them with several transformer blocks (Vaswani et al., 2017) using geometric transform attention (GTA) (Miyato et al., 2023). The feedforward layers, a two-layer MLP, operate directly on the geometric tokens' features, leaving their poses unchanged. As stated above, the features are canonicalized (e.g., expressed in the local frame determined by the token's pose) so the feedforward layers do not break $\mathrm{SE}(3)$ equivariance. The attention layers in our transformer use GTA, which modulates the similarity scores based on the relative transformation between tokens, using each token's $\mathrm{SE}(3)$ pose to compute the relative transformation. We use a combination of dot product similarity

(on the scalar and vector components) and negative Euclidean distance (on point components) to compute similarity. The dot product is, in our experience, more stable during training, but violates equivariance when used with point features. Since our model is used in the low data regime, we use negative Euclidean distance as needed to ensure exact $SE(3)$ equivariance out-of-the-box.

## 4.2 GEOMETRIC TOKEN ACTION DECODER

The output of our encoder network is a set of geometric tokens that form a 3D representation of the observed scene. To solve robotic manipulation tasks, we feed this representation to a decoder network that is trained with flow-matching to generate trajectories of $SE(3)$ gripper poses and states. We now describe a novel $SE(3)$ equivariant decoder network designed to output gripper actions in the relative frame. (We also describe how to produce actions in the absolute frame in Appendix F.)

With geometric tokens, the features remain canonicalized in some known reference frame throughout processing. This property can be used to our advantage to output end-effector trajectories in arbitrary reference frames. To predict actions relative to the end-effector frame, we create action geometric tokens where the pose is set to be the end-effector's pose at the current timestep. When these tokens are processed by a $SE(3)$ geometric transformer, the action geometric features are naturally suited to express the robot trajectory in the global frame while still having the precision of being described in the local frame.

In practice, we train an $SE(3)$ geometric transformer with a flow-matching loss to output the velocity field. The action geometric tokens for each timestep in the trajectory are initialized with the features sampled from a Normal distribution, and the poses are all set to the current gripper pose. The model is trained using the loss from Equation 4 applied to the predicted action token features only. We modify the $SE(3)$ geometric transformer architecture from our encoder to roughly match the transformer decoder from DiffPo. Each transformer block is composed of causal self-attention between action tokens, then cross-attention with the geometric tokens from the encoder, and finally a feedforward layer. We use four transformer blocks in total, and a final projection layer to convert he action geometric tokens into velocities. A diagram of the relative action decoder is shown on the right of Figure 1.

## 4.3 EQUIVARIANCE PROPERTIES

Our method is end-to-end equivariant with respect to global $SE(3)$ transformations of the robots, sensors, and objects. To simplify the analysis, let us consider a setup with two cameras and one gripper. The observation is composed of images, $I_1, I_2$, camera intrinsics, $K_1, K_2$, camera extrinsics, $M_1, M_2$, gripper pose, $P$, and gripper state, $\lambda$. The action of a global transformation, $g \in SE(3)$, on the inputs is: $I_1, I_2, K_1, K_2, M_1, M_2, P, \lambda \mapsto I_1, I_2, K_1, K_2, gM_1, gM_2, gP, \lambda$. In other words, the transformation only acts on the camera poses, $M_1, M_2$, and gripper pose, $P$. Our encoder network, call it $\psi$, outputs geometric tokens: $Z, G = \psi(I, K, M, P, \lambda)$, so the effect of a $g \in SE(3)$ is:

$$Z, gG = \psi(I_1, I_2, K_1, K_2, gM_1, gM_2, gP, \lambda), \tag{6}$$

meaning the output is transformed by an $SE(3)$ action on the geometric tokens' poses, with the tokens' features remaining unchanged. It is important to note that the equivariance does not capture independent camera transformations (we test generalization to this empirically in Section 5.1).

The equivariance of our method is slightly different from prior equivariant robot learning methods that operate on structure or spatial inputs (Wang et al., 2022; Yang et al., 2024) Those methods claim equivariance with respect to transformations of the robots and objects independent of the sensors, which results in stronger generalization. Equivariant methods that operate on RGB images, like RAVEN, cannot achieve this equivariance outside of a few edge cases (see longer discussion in Appendix A).

## 5 EXPERIMENTS

We systematically evaluate RAVEN in both simulation and real-world settings. Our experiments are designed to address the following research questions: (1) Does our method outperform existing approaches on general manipulation tasks? (2) Can it maintain strong performance under different

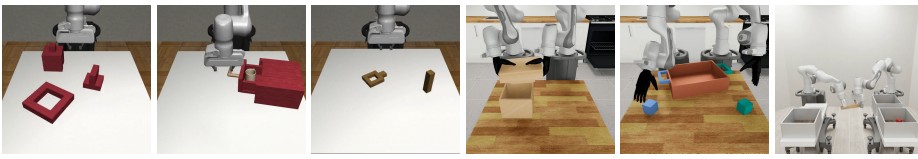

(a) Representative tasks from MimicGen (left three images) and DexMimicGen (right three images).

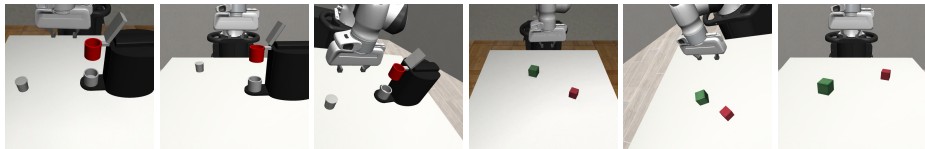

(b) Viewpoint generalization evaluation.

Figure 4: Visualization of tasks from the MimicGen and DexMimicGen benchmarks and the viewpoint generalization setting. Subfigure (a) shows representative tasks, while (b) illustrates viewpoint variation: in each task, the leftmost image is the original view and the two on the right are sampled perturbed views. The complete set of tasks is provided in Appendix D.

camera numbers and bimanual configurations? (3) Does it generalize to novel viewpoints at inference time? (4) Is its training speed faster than previous equivariant-based methods? (5) Is it applicable to real-world robotic manipulation tasks?

## 5.1 SIMULATION

**Experiment Setting** To address the first four questions, we conduct simulation experiments on the MimicGen (Mandlekar et al., 2023) and DexMimicGen (Jiang et al., 2024) benchmarks. First, we evaluate RAVEN on twelve single-arm manipulation tasks from MimicGen, a widely used suite for closed-loop policy learning (Wang et al., 2024; Hu et al., 2025), using agent-view and eye-in-hand RGB observations. Each model is trained with 100 demos. Second, to study multi-camera and bimanual settings, we evaluate RAVEN on six bimanual tasks from DexMimicGen, consisting of three gripper-based and three dexterous-hand tasks. This setting uses two eye-in-hand cameras and one agent-view, and involves controlling two arms with moving cameras, making the tasks more challenging. Models are trained with both 50 and 100 demos. Both MimicGen and DexMimicGen tasks are evaluated with 50 rollouts per task and 3 random seeds. Figure 4 (a) illustrates representative tasks from the two benchmarks. Third, to study viewpoint generalization, we modify the MimicGen tasks by randomly perturbing the agent-view camera at the start of each episode (and demonstration), as shown in Figure 4 (b). The camera is sampled within $\pm 20°$ pitch and $\pm 40°$ yaw relative to the original viewpoint, ensuring unique but distributionally consistent perspectives across demos and rollouts. Fourth, we compare the training speed of RAVEN with baselines to demonstrate its efficiency. Dataset configurations, visualizations of all subtasks, and training details are provided in Appendices D and E.

**Baselines** We compare against four strong baselines: **(1) DiffPo** (Chi et al., 2023): A diffusion-based policy without equivariance, trained from scratch, serving as the primary reference. **(2) DiffPo (Pre)**: The same as (1) but with a pretrained ResNet encoder to match the visual backbone used in RAVEN. This ensures a fair comparison when both methods leverage pretrained perception. **(3) EquiDiffPo** (Wang et al., 2024): An $SO(2)$-equivariant image-based diffusion model that incorporates both an equivariant image encoder and an equivariant temporal U-Net. **(4) ACT** (Zhao et al., 2023): A transformer-based behavior cloning method designed for closed-loop control. Additionally, we conduct experiments on each task to compare relative and absolute action prediction with RAVEN. Due to space constraints, the full results are provided in Appendix F.

**MimicGen** Table 1 reports the success rates across all methods and configurations. Compared to the strongest baseline result for each task, RAVEN achieves the highest performance on 11 out of 12 tasks, with the remaining task only 4% behind. These results demonstrate the robustness of our approach under different action representations. RAVEN exceeds the best-performing baseline

Table 1: MimicGen tasks with agent-view and eye-in-hand cameras using 100 demos. Results are reported as the best success rate (%) across 30 evaluation checkpoints during training, each based on the same set of 50 rollouts. The final score is averaged over the best checkpoint from each of 3 random seeds (some baselines from Wang et al. (2024)). Numbers in parentheses indicate the difference between RAVEN and the best baseline (improvements shown in blue, drops in red).

| Method | Average | Stack D1 | Stack Three D1 | Square D2 | Threading D2 | Coffee D2 | 3P Assembly D2 |
|---|---|---|---|---|---|---|---|
| RAVEN | 66 (+12) | 100 (+3) | 81 (+26) | 50 (+25) | 28 (+6) | 61 (+1) | 27 (+12) |
| EquiDiffPo | 54 | 93 | 55 | 25 | 22 | 60 | 15 |
| DiffPo (Pre) | 52 | 97 | 53 | 22 | 21 | 53 | 14 |
| DiffPo | 42 | 76 | 38 | 8 | 17 | 44 | 4 |
| ACT | 21 | 35 | 6 | 6 | 10 | 19 | 0 |
| | | H. Cleanup D1 | M. Cleanup D1 | Kitchen D1 | N. Assembly D0 | Pick Place D0 | Coffee Prep. D1 |
| RAVEN | | 73 (+8) | 53 (+4) | 85 (+5) | 74 (=) | 73 (+27) | 73 (-4) |
| EquiDiffPo | | 65 | 49 | 67 | 74 | 42 | 77 |
| DiffPo (Pre) | | 54 | 47 | 80 | 71 | 46 | 65 |
| DiffPo | | 52 | 43 | 67 | 55 | 35 | 65 |
| ACT | | 38 | 23 | 37 | 42 | 7 | 32 |

Table 2: Maximum success rate (%) on DexMimicGen tasks with 50 demos. Results are reported using the same evaluation protocol as in Table 1.

| Method | Average | Threading | 3P Assembly | Transport | B. Cleanup | D. Cleanup | Tray Lift |
|---|---|---|---|---|---|---|---|
| RAVEN | 82 (+17) | 63 (+34) | 69 (+19) | 76 (-5) | 98 (+13) | 97 (+1) | 90 (+34) |
| EquiDiffPo | 62 | 29 | 45 | 63 | 85 | 94 | 56 |
| DiffPo (Pre) | 65 | 25 | 50 | 81 | 85 | 96 | 55 |
| DiffPo | 60 | 20 | 43 | 78 | 84 | 91 | 43 |

by 12% on average. While both RAVEN and DiffPo (Pre) use a pretrained image encoder, RAVEN consistently outperforms it by 14%, indicating that our architectural design provides additional advantages beyond pretraining. Interestingly, DiffPo (Pre) performs on par with, or slightly worse than, EquiDiffPo on individual tasks, with an average gap of 2%, suggesting the positive effect of pretraining on model robustness. Nevertheless, all baselines still fall short by a substantial margin compared to RAVEN. Complete results, including standard deviations across three random seeds, are provided in Appendix G.

**DexMimicGen**  Table 2 reports results on DexMimicGen with 50 demos. Complete results with 100 demos and variance across seeds are provided in Appendix H. At the time of writing, only tasks with the D0 initial state distribution are publicly available, so there is minimal variation in object placement. Since generalization to spatial transformations is less useful in this benchmark, we expect the performance boost of equivariance to be less significant. Despite this, RAVEN consistently outperforms all baselines by a substantial margin. Notably, as shown in Table H, RAVEN trained with only 50 demos performs comparably to or even outperforms baselines trained with 100 demos, highlighting the superior data efficiency of our method. We believe our performance in this setting is driven by the GTA layers in the encoder, which allow information from all three cameras to be fused into a better understanding of the scene. In comparison, DiffPo and EquiDiffPo simply concatenate feature vectors from each image without reasoning over their geometric relationships, limiting the integration of 3D information.

**Viewpoint Generalization**  A key advantage of our observation encoder is its ability to form 3D representations, rather than relying solely on 2D image features. This inductive bias makes our method naturally more robust to variations in camera viewpoints compared to purely image-based approaches. This property is particularly useful in real-world deployment, where demos are often collected across a fleet of robots with varied sensors (Zhao et al., 2024). The results (Table 3) show that RAVEN is more robust to viewpoint perturbations than the baselines. Averaged across four tasks, RAVEN achieves 23% improvements over the strongest baseline. As expected, overall performance drops compared to Table 1 due to distribution shifts from unseen camera poses. It is worth noting that our method is equivariant to transformations of the *entire* scene, including robots, objects, and cameras, but this task perturbs only a single camera, which breaks global equivariance. Nonetheless,

Table 3: MimicGen experiments with randomly perturbed agent-view cameras. A subset of tasks is selected for evaluation. The same experimental protocol as in Table 1 is followed.

| | Average | Stack | Stack Three | Coffee | Hammer Cleanup |
|---|---|---|---|---|---|
| RAVEN | 71 (+23) | 99 (+21) | 68 (+26) | 53 (+2) | 65 (+13) |
| EquiDiffPo | 48 | 65 | 25 | 51 | 52 |
| DiffPo (Pre) | 44 | 78 | 42 | 21 | 36 |

Table 4: Ablation Study of RAVEN on MimicGen tasks, success rate (%) averaged over three seeds.

| | Average | Stack | Stack Three | Coffee | Hammer Cleanup |
|---|---|---|---|---|---|
| RAVEN | 76 | 100 | 81 | 50 | 73 |
| w/o SE(3) ray encoding | 72 | 100 | 74 | 44 | 69 |
| w/o Equi. Decoder | 72 | 99 | 86 | 38 | 65 |
| w/o Equi. Encoder & Decoder | 58 | 90 | 56 | 27 | 60 |

our geometry-aware feature representations enable significantly better generalization to novel views than all baselines.

We also performed an experiment to test RAVEN's generalization to noisy camera calibration (see Appendix B.1). We find that RAVEN is sensitive to noise in the camera calibration, especially when introduced at test time. However, by adding data augmentation to the camera parameters during testing, RAVEN is robust to calibration errors, dropping only 1% in success rate averaged across a subset of MimicGen tasks.

**Ablation Study**    We perform an ablation study to determine the impact of RAVEN's equivariant layers and geometric structure have on performance. We consider three variations of our method. First, we remove the SE(3) ray encoding scheme and instead encode ray patches as elements of ray-space, $\mathbb{R}^3 \times S^2$ (e.g. each ray patch has a direction vector instead of an orientation). Second, we remove the equivariant layers in the decoder network by replacing all GTA layers with standard dot-product attention. We still provide the network with the pose of all tokens via an absolute position embedding. Third, we remove the equivariant layers in the decoder and encoder in the same manner.

For each variation, we evaluate its performance on a subset of MimicGen tasks and report success rates in Table 4. Without SE(3) ray encoding, the performance drops by 4% on average. While this drop is small, it highlights the benefits of the additional geometric structure provided by the ray patches orientation. The method's performance drops an average 4% with a non-equivariant decoder and 18% with a non-equivariant encoder and decoder. This supports the importance of equivariant layers in our method while keeping the added geometric structure of encoding images as rays. These tasks are all table-top tasks, with minimal out-of-plane motions and we expect the impact of SE(3) equivariant layers to be greater in less structured manipulation settings.

**Policy Efficiency Analysis**    Training and deploying policies efficiently is critical in robotics applications. In this context, training time is an important evaluation metric. Prior equivariant methods often suffer from longer training durations (Wang et al., 2024; Hu et al., 2025). To evaluate training efficiency, we measure the training time on the MimicGen *Threading D2* task under identical hardware and configurations. DiffPo completes the pipeline (training plus policy rollout in simulation) in 3.3 hours, while EquiDiffPo requires 7.3 hours. In contrast, RAVEN finishes in just **2.8** hours. This corresponds to a **1.6×** speedup over EquiDiffPo with a superior performance, highlighting our method's advantages in both effectiveness and training efficiency.

## 5.2    REAL WORLD

**Physical Setup**    Our real-world experiments are conducted using a Universal Robot UR5 equipped with a Robotiq-85 Gripper and custom-designed soft fingers. We use one side-mounted RealSense D455i for agent-view observations and a wrist-mounted GoPro for eye-in-hand views. Demos are collected via the Gello teleoperation interface (Wu et al., 2024), with RGB observations and actions recorded at 5 Hz. Figure 5 shows the four manipulation tasks. *Banana Picking*, *Beans Scooping*, and

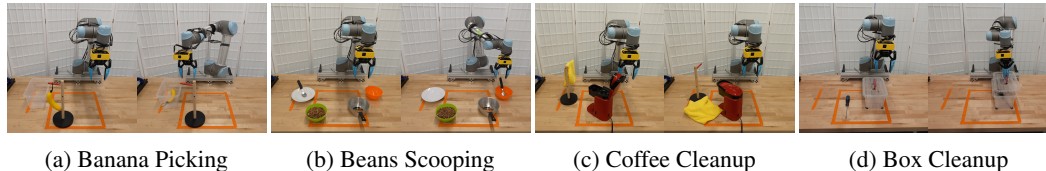

| (a) Banana Picking | (b) Beans Scooping | (c) Coffee Cleanup | (d) Box Cleanup |

Figure 5: Visualization of four real world tasks shown at the beginning (left) and end (right) of the episode. The randomness of task initial states is illustrated in Fig. 11.

Table 5: Real-world task performance over 20 trials, reported as average progress (%) and, in parentheses, success rate (%).

|  | Banana Picking | Beans Scooping | Coffee Cleanup | Box Cleanup |
|---|---|---|---|---|
| # Demos | 75 | 75 | 75 | 40 |
| RAVEN | 100 (100) | 70 (45) | 75 (45) | 77 (60) |
| DiffPo (Pre) | 53 (50) | 30 (0) | 56 (15) | 43 (30) |

*Coffee Cleanup* require precise 3D control to meet complex goal conditions, presenting challenges in spatial reasoning. The final task, *Box Cleanup*, uses only half the demos compared to other tasks. It emphasizes learning under limited data and involves transparent objects that require policies to adapt to low-data regimes while interpreting challenging visual cues from RGB observations. We compare RAVEN against the pretrained Diffusion Policy (Chi et al., 2023). Further details, including hardware setup, task specifications, and success rate definition, are provided in Appendix I.

**Results**    We report the results of four real-world tasks in Table 5. Numbers represent the average completion progress of each task, with overall task success rate shown in parentheses. Progress evaluation offers a more comprehensive measure of policy performance, particularly for tasks where baselines fail entirely (e.g., 0% in *Beans Scooping*). Across all tasks, our method consistently outperforms the pretrained DiffPo. Specifically, RAVEN achieves 81% in average progress and 63% in overall completion success rate. By contrast, DiffPo (Pre) reaches only 46% and 24%. These results highlight the strong effectiveness of our method, particularly its data efficiency, and demonstrate its practicality for real-world deployment. Progress definitions and full results with our additional absolute action variant are provided in Appendix I.

## 6    CONCLUSION

In the space of Imitation Learning methods for robotic manipulation, the RAVEN framework is an exciting new piece of technology that combines the advantages of standard imitation learning methods (like baseline Diffusion Policy) with the advantages of an equivariant model. Unlike previous equivariant models like EquiDiffPo (Wang et al., 2024) or EquiBot (Yang et al., 2024), which require depth or point cloud data in order to be SE(2)- or SE(3)-equivariant, RAVEN is an end-to-end SE(3)-equivariant method that can function with RGB input alone. Moreover, RAVEN can benefit from standard pretrained image encoders, just like non-equivariant models do. Most importantly, compared to a baseline pretrained Diffusion Policy, RAVEN outperforms significantly and has faster training times.

**Limitations**    RAVEN has some limitations. Although a RAVEN policy can generalize well to novel camera viewpoints, it requires all cameras to be well calibrated with respect to a world coordinate frame. Also, we have so far only explored this idea in the context of RGB inputs. In principle, the RAVEN encoder is compatible with a variety of input modalities, including depth and point cloud data, and integrating with this type of information is something we would like to explore in the future.

## ACKNOWLEDGEMENTS

We would like to thank Haotian Liu for his assistance with the real-world experiments as well as all members of the Geometric Learning Lab and Helping Hands Lab for their valuable feedback on the manuscript. This work was supported in part by NSF Grants 2107256, 2134178, 2314182, 2409351, 2442658 and NASA Grant 80NSSC19K1474. This work used GPU resources at NCSA Delta through allocation CIS250208 from the Advanced Cyberinfrastructure Coordination Ecosystem: Services & Support (ACCESS) program (Boerner et al., 2023), which is supported by U.S. National Science Foundation grants #2138259, #2138286, #2138307, #2137603, and #2138296.

## REPRODUCIBILITY STATEMENT

Comprehensive experimental setups and training details are provided in the Appendix. Upon acceptance, we will release our implementation publicly via a GitHub repository.

## ETHICS STATEMENT

This research makes use of both publicly available datasets, released under appropriate licensing agreements, and additional data collected by the authors. All newly collected data involve only robotic systems and physical objects, without any human participants or personally identifiable information. No sensitive or harmful content is present. Data collection and usage were conducted solely for scholarly research purposes and in accordance with established ethical standards. Our work conforms, in every respect, with the ICLR Code of Ethics.

## THE USE OF LARGE LANGUAGE MODELS (LLMS)

LLMs were used exclusively for improving language quality, such as grammar correction and refinement. The core method development in this research does not involve LLMs as any important, original, or non-standard components.

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

## A    EXTENDED EQUIVARIANCE DISCUSSION

In this section, we continue the discussion from Section 4.3 about how the equivariance of RAVEN differs from the traditionally accepted definition in equivariant robot learning. To restate the distinction, our method's equivariance is with respect to joint transformations of the robots, objects and sensors whereas traditionally it is with respect to joint transformations of the robots and objects (not the sensors). We will refer to these two definitions of equivariance as *robot-object-sensor equivariance* and *robot-object equivariance*, respectively. It is clear to see that the robot-object-sensor equivariance of our method is less powerful than robot-object equivariance, since there is only generalization if the sensors move with the robots and objects. The reason our method cannot achieve robot-object equivariance comes down to the input data. With RGB images, there is not an obvious mapping on the input space that captures a transformation of scene independent of the camera. (In fact, there is an entire field of novel view synthesis that trains large networks to find such a mapping). In contrast, for spatial inputs like point clouds, we can apply $4 \times 4$ transformation matrix to all points to approximate how the transformed scene would appear independent of camera pose. In practice, the gap between our method's equivariance and robot-object equivariance is not always obvious or even present, as we show in the special cases below.

$SE(2)$ **Equivariance with a Single RGB Camera:**    With a single camera, 2D roto-translations of the image correspond to 2D roto-translations of the scene (i.e. robots and objects) about the cameras roll axis, independent of the camera pose. This is the setting explored by (Wang et al., 2022; 2024). RAVEN can match the robot-object equivariance of these methods here, so long as an 2D equivariant convolutional encoder network is used.

**Discretized** $SO(2)$ **Equivariance with a Structured RGB Camera Layout**    If cameras are arranged uniformly around the scene's z-axis, then the group action takes the form of permuting the images between cameras. For instance, if four cameras are arranged 90 degrees apart, then cycling the images between cameras corresponds to rotations of the scene (robots and objects) by 90-degree increments. In this setting, RAVEN would be robot-object equivariant to a discrete subgroup of $\mathrm{SO}(2)$ (or subgroups of $\mathrm{SO}(3)$ using camera layout shown in (Esteves et al., 2019)).

$SE(3)$ **Equivariance with Partial Point Clouds**    When only a few depth sensors are used, the point cloud will often have some gaps due to occlusion. To be precise, applying 3D roto-translations to the points corresponds to joint transformations of the robots, objects and sensors (since the occluded regions also transform). In this case, methods can only achieve robot-object-sensor equivariance. This happens to be the case in several equivariant point cloud works Yang et al. (2024); Ryu et al. (2022; 2024), although they do not discuss the nuance of their equivariance claims. So unless many depth sensors are used, our method will have the same equivariance guarantees. In practice, point cloud networks seem to generalize well on partial point clouds and this distinction is worth exploring more in the future.

## B    ADDITIONAL EXPERIMENTS

### B.1    ROBUSTNESS TO CAMERA CALIBRATION ERROR

A key assumption of RAVEN is that the camera intrinsic and extrinsics are known during training and deployment. Slight errors in camera parameters are typical in real robotic systems, due to miscalibration or sensors being bumped. Since RAVEN uses the camera parameters to map features on 2D image plane to SE(3) ray features, it is possible that these errors would harm RAVEN's ability to form 3D representations of the scene and accurately predict actions. In this experiment, we directly test the effect of noisy camera parameters on RAVEN's performance.

We look at two cases of camera calibration error. In the first case, the model is trained with ground-truth camera parameters but tested with noisy camera parameters. This case may arise in practice when using a fleet of robots. While the demonstration data may be generated on a carefully calibrated robot, other robots may fall out of calibration during deployment. In the second case, the model is trained and tested with the same noisy parameters. This case represents a scenario where the method is used on one robot, but the calibration is done quickly and, therefore, is inaccurate. For

these two cases, we evaluate the performance of RAVEN on a subset of MimicGen tasks, following same protocol as in Table 1. The ground-truth camera parameters are provided by the simulation environment. For noisy parameters, we sample some noise and add it as an offset to the ground-truth parameters. Specifically, we apply a ±5 pixel (6.6%) shift to the principal point, a 2% variation in focal length, a 0.5 to 1.5 cm perturbation to the camera translation, and a 6, -4, and 8 degree variation to the camera roll, pitch and yaw angles. The results, shown in Table 6,

Lastly, we look at using data augmentation to improve robustness to camera calibration error. In the final row of Table 6, we include a comparison of RAVEN that was trained with data augmentation on the ground truth camera parameters (both intrinsic and extrinsic), and tested on noisy camera parameters. With data augmentation, the impact of noisy camera parameters is minimal, lowering performance by 3% on Coffee task and 0% on the other tasks. This experiment shows that RAVEN is robust to reasonable camera calibration errors, so long as data augmentation is used.

Table 6: Effect of feature map grid resolution on RAVEN's performance. Performance reported in success rate (%) on subset of MimicGen tasks, averaged over three seeds.

| Training Parameters | Testing Parameters | Stack | Stack Three | Coffee | Hammer Cleanup |
|---|---|---|---|---|---|
| ground truth | ground truth | 100 | 81 | 50 | 73 |
| noisy | noisy | 100 | 80 | 49 | 71 |
| ground truth | noisy | 100 | 76 | 42 | 69 |
| ground truth + data aug | noisy | 100 | 81 | 47 | 73 |

## B.2 COMPARING GRID RESOLUTIONS FOR GENERATING RAYS

As stated in Section 4.1, we use a pretrained ResNet encoder to preprocess image observations into feature maps. Each element in the feature map is then mapped to an $SE(3)$ ray encoding. Since the ResNet downsamples the image, we end up with much fewer ray tokens than image pixels, which accelerates subsequent attention layers (which are quadratic in number of tokens). In this experiment, we compare performance when using different feature map grid resolutions. Specifically, we test with 3x3 feature maps, 5x5 feature maps (this is the default), and 7x7 feature maps. We vary the resolution by removing maxpool layers or reducing convolutional stride in the ResNet; all other model parameters are kept the same. We report the success rates on a subset of MimicGen tasks in Table 7.

The results show that a 5x5 grid resolution performs best. There is a larger drop-off in performance with a 3x3 grid than a 7x7 grid. We believe a higher grid resolution allows the model to construct a denser 3D understanding of the scene. It is worth pointing out that downsampling in the image encoder does not affect the $SE(3)$ equivariance of our method, even though it does affect equivariance to image shifts. This is because the $SE(3)$ transformation acts on the camera extrinsincs, not the image. In future work, it may be possible to achieve equivariance to image shifts (as in shift an image to model the effect of translating the camera position). This could further increase robustness to camera viewpoint changes.

Table 7: Effect of feature map grid resolution on RAVEN's performance. Performance reported in Success Rate (%) on subset of MimicGen tasks, averaged over three seeds.

| | Stack | Stack Three | Coffee | Hammer Cleanup |
|---|---|---|---|---|
| RAVEN (3x3 grid) | 99 | 73 | 37 | 67 |
| RAVEN (5x5 grid) | 100 | 81 | 50 | 73 |
| RAVEN (7x7 grid) | 100 | 79 | 46 | 74 |

## B.3 FLOW MATCHING VS. DIFFUSION

In this experiment, we compare flow matching and diffusion for denoising action predictions. We compare performance on a subset of MimicGen tasks in Table 8. We kept the model architecture the same, and only changed the loss applied and the denoising scheme at inference. For flow-matching,

we followed the procedure and parameters presented in this work. For diffusion, we used the approach from Wang et al. (2024). The results show that there is a small performance boost for flow-matching, although it varies by task. Based on our experience developing this work, diffusion can sometimes perform better on tasks that require high precision. We did not focus any efforts on refining the parameters for the denoising schemes, and it is possible that additional improvements to performance and speed could be realized after fine tuning.

Table 8: Comparing performance of RAVEN using Flow-Matching and Diffusion for decoding actions. Performance reported in Success Rate (%) on subset of MimicGen tasks, averaged over three seeds.

|  | Stack | Stack Three | Coffee | Hammer Cleanup |
|---|---|---|---|---|
| RAVEN (Flow Matching) | 100 | 81 | 50 | 73 |
| RAVEN (Diffusion) | 100 | 84 | 43 | 70 |

## C  GEOMETRIC TOKENS FOR OTHER OBSERVATION MODALITIES

In this work, we focus on observations with multiple images and the end-effector state. We believe that our method can be easily extended to robotic settings with other observation modalities. Integrating other inputs comes down to converting the data into geometric tokens. In most cases, the token features are generated with a linear layer or a traditional network encoder (ViT or PointNet). The token poses may be less obvious, and we provide some suggestions here. We would like to empirically test these ideas in the future.

**Joint Data**    In some tasks, we may wish to encode the current joint positions of the robot, especially for joint space action,s as in ALOHA (Zhao et al., 2024). One option is to convert the joint angles to features with a linear layer and assign the pose to be the robot base frame. Another option is to use forward kinematics to generate a point cloud that represents the positions or meshes of each link, which exist in the robot base frame.

**Torque/Force Data**    This data already has a geometric structure, so a hard-coded mapping to vector and point features could be used. It would make sense to pose the data in the end-effector frame (for wrist-mounted force sensor) or a joint frame for motor torques.

**Point Cloud Data**    A point cloud cannot be canonicalized in the same way that an image or end effector can. So, our recommendation is to patchify the point cloud using clustering, embed each cluster into geometric features using PointNet, and assign the pose based on the cluster centroid and principal axes.

## D  SIMULATION EXPERIMENT DETAILS

We visualize the manipulation tasks from MimicGen (Mandlekar et al., 2023), DexMimicGen (Jiang et al., 2024), and the Viewpoint Generalization task in Figure 6, 7, and 8, respectively. Following prior work (Chi et al., 2023; Wang et al., 2024; Hu et al., 2025), we set the resolution of the image to $3 \times 84 \times 84$ and adopt the same maximum episode length. In addition, we slightly modify the original simulator so that the environments can provide intrinsic and extrinsic camera parameters in real time. For the viewpoint generalization task, we select a subset of tasks whose scene content is preserved under camera placement changes, ensuring that varying the viewpoint does not lead to significant information loss in the observations.

## E  TRAINING DETAILS

To capture temporal context, we follow prior work (Chi et al., 2023; Wang et al., 2024) and use a two-step history window on all observations as input to the policy. Images are encoded by a pretrained ResNet-18 (He et al., 2016) which was modified by replacing batchnorm with groupnorm layers (as done in DiffPo) and tweaking downsampling operations to produce a dense feature map ($5 \times 5$ grid

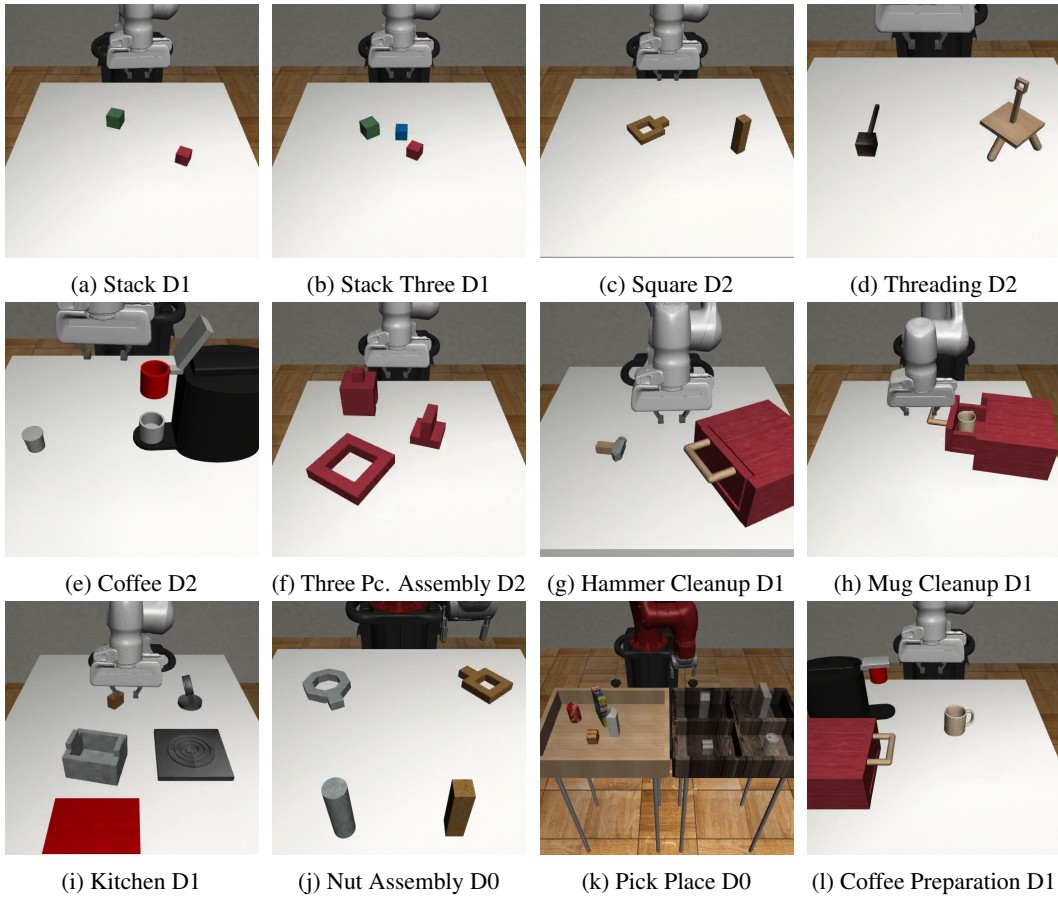

(a) Stack D1   (b) Stack Three D1   (c) Square D2   (d) Threading D2

(e) Coffee D2   (f) Three Pc. Assembly D2   (g) Hammer Cleanup D1   (h) Mug Cleanup D1

(i) Kitchen D1   (j) Nut Assembly D0   (k) Pick Place D0   (l) Coffee Preparation D1

Figure 6: The twelve simulation tasks from the MimicGen (Jiang et al., 2024) benchmark, each shown in its initial configuration.

cells from $84 \times 84$ simulation images; $7 \times 7$ grid cells with real-world images). The feature vector at each grid cell is mapped to the geometric token representation using a linear projection. For all geometric tokens, we set the dimensionality of the scalar, vector, and point features to be 64 (so there are $64 + 64 * 3 + 64 * 3$ channels). The pose of each grid cell is calculated using camera projection equations (Equation 2 shows the case for the z-direction, and we project the image plane axis as well to get the x- and y-directions). Our $\mathrm{SE}(3)$ Geometric Transformer is composed of two transformer blocks that use multi-headed attention (8 heads).

These geometric tokens produced by the encoder are fed to a decoder that either (i) uses EquiBot (Yang et al., 2024) to predict an absolute trajectory or (ii) our GTA Transformer to predict a relative trajectory. The decoder outputs a sequence of 16 action steps; all steps supervise training, while only the first 8 steps are executed at evaluation time. During training, we apply random cropping to a fixed resolution; at evaluation, we use center cropping. In simulation, the observation size is $84 \times 84$, whereas for real-world experiments, we found this resolution insufficient for fine-grained manipulation and therefore use $224 \times 224$. We train with AdamW (Loshchilov & Hutter, 2017) and exponential moving average (EMA). The hyperparameters for flow matching were based on Chisari et al. (2024): 10 integration steps for training and testing, a position embedding scale of 20.0 (for time step encoding), a noise scaling of 1.0, a linear flow schedule, an exponential scaling of 4.0, and a uniform SNR sampling scheme, and loss weights of 10 for end-effector rotation error, 10 for translation error, and 1 for the gripper state error. For all baselines, we keep their original hyperparameters and only align the number of training steps for a fair comparison. All runs use one GPU per experiment, executed on compute clusters and workstations equipped with multiple high-performance consumer-grade GPUs.

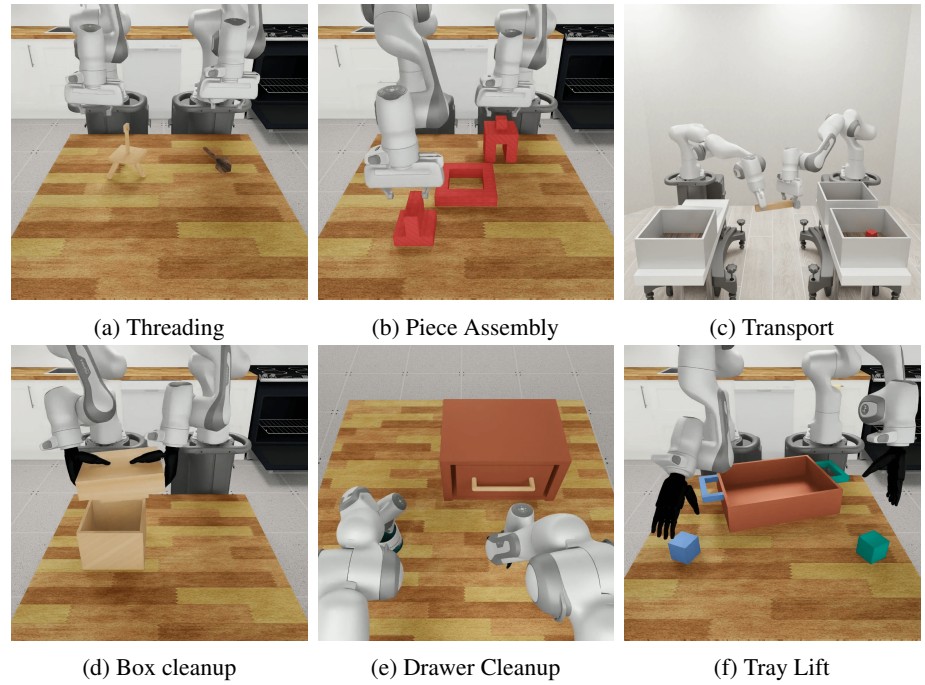

(a) Threading   (b) Piece Assembly   (c) Transport

(d) Box cleanup   (e) Drawer Cleanup   (f) Tray Lift

Figure 7: The six simulation tasks from the DexMimicGen (Jiang et al., 2024) benchmark are shown, with the top row ((a)-(c)) showing gripper-based tasks and the bottom row ((d)-(e)) illustrating dexterous hand-based tasks.

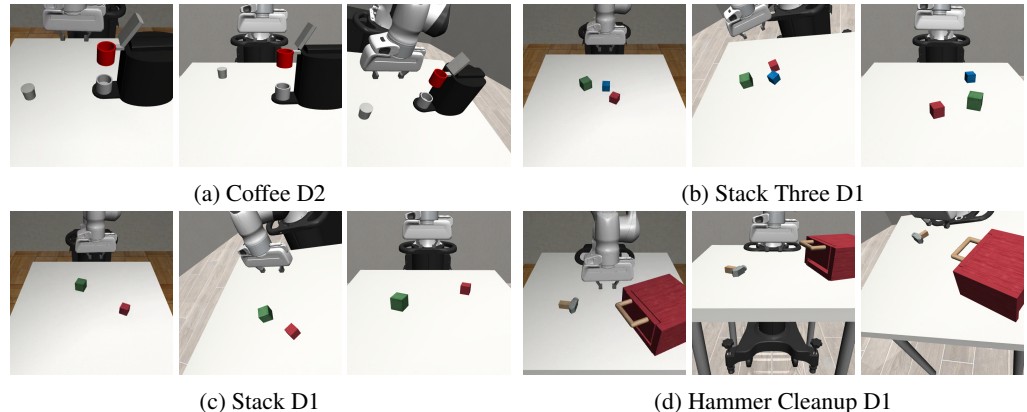

(a) Coffee D2   (b) Stack Three D1

(c) Stack D1   (d) Hammer Cleanup D1

Figure 8: Viewpoint generalization experiments on four MimicGen tasks under perturbed camera views.

## F    RAVEN WITH ABSOLUTE ACTION PREDICTION

In this section, we introduce a variation of RAVEN that employs a different action decoder to predict actions in the absolute frame. We refer to this variant as RAVEN-abs. The action decoder in RAVEN-abs follows the EquiBot (Yang et al., 2024) design: a 1D conditional UNet in which all linear layers are made SO(3)-equivariant using Vector Neurons (Deng et al., 2021). Translational equivariance is achieved through canonicalization, where a predicted offset vector is subtracted from the UNet inputs and added back to the predicted action at the output. Adapting the encoder outputs of RAVEN, a set of geometric tokens, to EquiBot's UNet requires several preprocessing steps. First, each token is decanonicalized by transforming it into the global frame, after which all tokens are pooled into a single feature vector. The offset vector is then obtained by mean-pooling the point components of

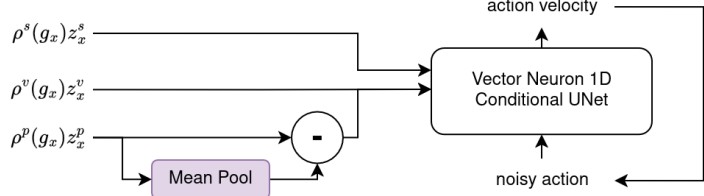

Figure 9: Process of feeding geometric tokens from our encoder network to the Unet Decoder from Equibot.

Table 9: Maximum success rates (%) on MimicGen Tasks with 100 demos.

| Method | Average | Stack D1 | Stack Three D1 | Square D2 | Threading D2 | Coffee D2 | 3P Assembly D2 |
|---|---|---|---|---|---|---|---|
| RAVEN | 66 | 100 | 81 | 50 | 28 | 61 | 27 |
| RAVEN-abs | 65 | 100 | 77 | 47 | 31 | 63 | 41 |

| | H. Cleanup D1 | M. Cleanup D1 | Kitchen D1 | N. Assembly D0 | Pick Place D0 | Coffee Prep. D1 |
|---|---|---|---|---|---|---|
| RAVEN | 73 | 53 | 85 | 74 | 73 | 73 |
| RAVEN-abs | 63 | 54 | 79 | 88 | 60 | 87 |

this vector. The scalar components are used as scalar conditioning for the UNet, while the vector components and the centered point components provide vector conditioning. This overall pipeline is illustrated in Figure 9.

We conduct experiments comparing the absolute action decoding scheme with the originally proposed relative decoding approach on MimicGen (Table 9) and DexMimicGen (Table 10) tasks, using the same experimental settings as in the main paper. The results show that RAVEN-abs performs comparably to RAVEN in single-arm manipulation tasks but underperforms in bimanual settings. This gap is expected and can be attributed to the increased complexity of the bimanual action space. In such settings, absolute trajectory representations may lead to larger discrepancies between actions. For instance, when the two arms consistently move in opposite directions, their translation values differ significantly. In contrast, relative trajectories remain more stable and consistent across arms, potentially making them easier to model.

## G  FULL MIMICGEN RESULTS WITH STANDARD ERRORS

Table 11 shows the MimicGen benchmark results corresponding to Table 1, but with the standard error included.

## H  FULL DEXMIMICGEN RESULTS WITH STANDARD ERRORS

Table 12 shows the DexMimicGen benchmark results corresponding to Table 2, but with the standard error included.

## I  REAL WORLD EXPERIMENT DETAILS

**Hardware Setup**  Figure 10 illustrates our real-world experimental setup. Two calibrated cameras (one GoPro for the eye-in-hand view and one RealSense D455i for the agent-view) are used to capture the scene, both aligned to a common global frame with their intrinsics and extrinsics recorded. Demos are collected using the Gello teleoperation interface (Wu et al., 2024) within a 40cm × 40cm square area, outlined in orange. At each timestep, we synchronously record visual observations together with end-effector actions, including position, rotation, and gripper state.

**Task Specifications**  Figure 11 shows averaged visualizations across multiple initial state distributions for each task. In *Banana Picking*, the mug tree is translated randomly within the square area, and its orientation is varied ±45° when the branch is aligned horizontally to the robot base square. In

Table 10: Maximum success rate (%) on DexMimicGen tasks with 50 demos.

| Method | **Average** | Threading | 3P Assembly | Transport | B. Cleanup | D. Cleanup | Tray Lift |
|---|---|---|---|---|---|---|---|
| RAVEN | 82 | 63 | 69 | 76 | 98 | 97 | 90 |
| RAVEN-abs | 77 | 47 | 55 | 91 | 86 | 97 | 83 |

Table 11: We report maximum success rates (%) on MimicGen tasks with 100 demos, averaged over three random seeds across different methods. Standard errors are indicated by $\pm$.

| Method | Stack D1 | Stack Three D1 | Square D2 | Threading D2 | Coffee D2 | Three Piece Assembly D2 |
|---|---|---|---|---|---|---|
| RAVEN | $100.0 \pm 0.0$ | $80.7 \pm 1.2$ | $50.0 \pm 4.0$ | $28.0 \pm 2.0$ | $60.7 \pm 5.0$ | $26.7 \pm 4.2$ |
| RAVEN-abs | $100.0 \pm 0.0$ | $76.7 \pm 4.6$ | $47.3 \pm 6.1$ | $31.3 \pm 7.0$ | $63.3 \pm 3.1$ | $40.7 \pm 3.1$ |
| EquiDiffPo | $93.3 \pm 0.7$ | $54.7 \pm 5.2$ | $25.3 \pm 8.7$ | $22.0 \pm 1.2$ | $60.0 \pm 2.0$ | $15.3 \pm 1.8$ |
| DifPo(Pre) | $96.7 \pm 1.2$ | $52.7 \pm 3.1$ | $22.0 \pm 2.0$ | $20.7 \pm 1.2$ | $52.7 \pm 2.3$ | $14.0 \pm 5.3$ |
| DiffPo | $76.0 \pm 4.0$ | $38.0 \pm 0.0$ | $8.0 \pm 1.2$ | $17.3 \pm 1.8$ | $44.0 \pm 1.2$ | $15.3 \pm 1.8$ |
| ACT | $34.7 \pm 0.7$ | $6.0 \pm 2.3$ | $6.0 \pm 0.0$ | $10.0 \pm 1.2$ | $19.3 \pm 2.4$ | $0.0 \pm 0.0$ |

| | Hammer Cleanup D1 | Mug Cleanup D1 | Kitchen D1 | Nut Assembly D0 | Pick Place D0 | Coffee Preparation D1 |
|---|---|---|---|---|---|---|
| RAVEN | $72.7 \pm 3.1$ | $53.5 \pm 2.3$ | $84.7 \pm 1.2$ | $74.0 \pm 5.3$ | $73.0 \pm 4.6$ | $72.7 \pm 4.2$ |
| RAVEN-abs | $62.7 \pm 1.2$ | $54.0 \pm 3.5$ | $79.3 \pm 4.2$ | $87.7 \pm 3.1$ | $60.3 \pm 3.1$ | $87.3 \pm 6.1$ |
| EquiDiffPo | $65.3 \pm 0.7$ | $49.3 \pm 0.7$ | $67.3 \pm 0.7$ | $74.0 \pm 1.2$ | $41.7 \pm 3.2$ | $76.7 \pm 0.7$ |
| DiffPo(Pre) | $54.0 \pm 2.0$ | $46.7 \pm 5.0$ | $80.0 \pm 2.0$ | $71.0 \pm 2.0$ | $46.3 \pm 4.7$ | $65.3 \pm 1.2$ |
| DiffPo | $52.0 \pm 1.2$ | $42.7 \pm 0.7$ | $66.7 \pm 2.4$ | $54.7 \pm 2.3$ | $35.3 \pm 2.2$ | $65.3 \pm 0.7$ |
| ACT | $38.0 \pm 4.2$ | $23.3 \pm 0.7$ | $37.3 \pm 3.5$ | $42.3 \pm 2.9$ | $7.2 \pm 0.9$ | $32.0 \pm 2.0$ |

*Beans Scooping*, the spoon is initialized with three discrete orientations, while the bean-filled bowl and the pot are randomly placed within the square area. In *Coffee Cleanup*, the rag is hung on the mug tree with the branch orientation randomized $\pm 20°$. The coffee machine is randomly placed within the square area, and its orientation is additionally perturbed by up to $\pm 20°$ when positioned $45°$ off perpendicular to the robot base square. In *Box Cleanup*, a screwdriver is randomly placed within the square area, with its head pointing outward perpendicular to the robot base and perturbed by up to $\pm 60°$. We also visualize one episode for each task in Figure 12, where the subfigures illustrate the key frames corresponding to critical action steps.

**Progress and Subgoal Definitions** In our evaluation, rather than using only a binary final outcome (0 for failure and 1 for success), we also introduce a progress reward to better capture the policy's performance. Specifically, we define a sequence of key subgoals for each task, evenly distributing weights across them such that the total sums to 1. The progress reward is then computed by accumulating the weights of completed key subgoals:

- *Banana Picking*:
  - Grasp and lift the banana
  - Drop the banana into the transparent box

- *Beans Scooping*:
  - Grasp the spoon
  - Scoop the beans
  - Transfer beans into the pot
  - Place the spoon back into the bowl

- *Coffee Cleanup*:
  - Grasp the rag from the mug tree
  - Wipe the coffee machine
  - Close the machine cover
  - Press the handle to lock the cover

- *Box Cleanup*:
  - Remove the box cover

Table 12: We report maximum success rates (%) on DexMimicGen tasks with 50 and 100 demos, averaged over three random seeds across different methods. Standard errors are indicated by $\pm$.

| Method | Threading 50 | Threading 100 | Piece Assembly 50 | Piece Assembly 100 | Transport 50 | Transport 100 | Avg (50 demos) |
|---|---|---|---|---|---|---|---|
| RAVEN | $63.3 \pm 3.1$ | $80.0 \pm 2.0$ | $69.3 \pm 6.1$ | $78.0 \pm 7.2$ | $76.0 \pm 4.0$ | $80.0 \pm 6.0$ | 82.2 |
| RAVEN-abs | $46.7 \pm 1.2$ | $75.3 \pm 5.0$ | $54.7 \pm 2.3$ | $76.0 \pm 7.2$ | $91.3 \pm 3.1$ | $82.0 \pm 2.0$ | 76.9 |
| EquiDiffPo | $28.7 \pm 3.1$ | - | $45.3 \pm 4.6$ | - | $62.6 \pm 2.3$ | - | 62.0 |
| DiffPo (Pre) | $25.3 \pm 4.2$ | $54.7 \pm 3.1$ | $50.0 \pm 3.5$ | $64.7 \pm 8.3$ | $80.7 \pm 2.3$ | $72.7 \pm 9.9$ | 65.3 |
| DiffPo | $20.0 \pm 2.0$ | $42.7 \pm 4.2$ | $42.7 \pm 6.4$ | $71.3 \pm 6.4$ | $78.0 \pm 3.5$ | $76.6 \pm 3.1$ | 59.8 |

| Method | Box Cleanup 50 | Box Cleanup 100 | Drawer Cleanup 50 | Drawer Cleanup 100 | Tray Lift 50 | Tray Lift 100 | Avg (100 demos) |
|---|---|---|---|---|---|---|---|
| RAVEN | $98.0 \pm 0.0$ | $96.7 \pm 1.2$ | $96.7 \pm 1.2$ | $99.3 \pm 1.2$ | $90.0 \pm 6.9$ | $91.3 \pm 1.2$ | 87.6 |
| RAVEN-abs | $88.0 \pm 2.0$ | $84.7 \pm 1.2$ | $97.3 \pm 1.2$ | $95.3 \pm 2.3$ | $83.3 \pm 4.2$ | $87.2 \pm 2.3$ | 83.4 |
| EquiDiffPo | $85.3 \pm 4.2$ | - | $94.0 \pm 5.3$ | - | $56.0 \pm 7.2$ | - | - |
| DiffPo (Pre) | $85.3 \pm 3.1$ | $88.7 \pm 5.8$ | $96.0 \pm 3.5$ | $98.7 \pm 1.2$ | $54.7 \pm 6.4$ | $80.0 \pm 7.2$ | 76.6 |
| DiffPo | $84.0 \pm 3.5$ | $88.7 \pm 3.1$ | $91.3 \pm 3.1$ | $92.7 \pm 2.3$ | $42.7 \pm 2.3$ | $71.3 \pm 6.1$ | 73.9 |

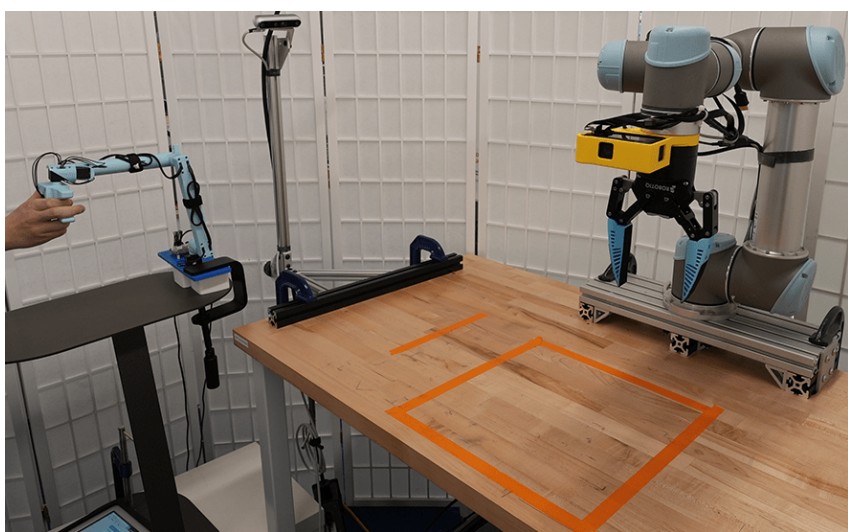

Figure 10: The real-robot platform consists of a UR5 robot arm equipped with a GoPro for the eye-in-hand view and an Intel RealSense D455i camera mounted on the left as the agent camera. The Gello teleoperation interface is also shown on the left.

– Place the screwdriver into the box
– Put the cover back on

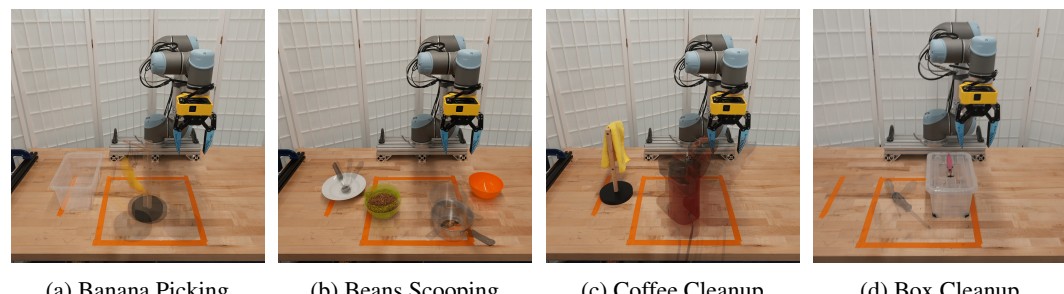

(a) Banana Picking    (b) Beans Scooping    (c) Coffee Cleanup    (d) Box Cleanup

Figure 11: Distributions of random initial states used in the four real-world tasks.

**Full Real-world Results**   Table 13 reports the full results of the four real-world experiments with the additional absolution action representation variant, including subgoal-level success rate (%), average progress reward (%), and overall task completion rate (%).

**Discussion of RAVEN's Failure Modes**   In the beans scooping task, the most common failure was the robot getting stuck repeating the same dropping action sequence (e.g. tilting scooper) even though the beans were dropped already. In the coffee cleanup task, the failures were either getting stuck in the wiping stage or inaccurately pressing the handle when closing the lid. The demonstrations included three wiping motions in a row, so it makes sense that the robot would get stuck here since it has limited observation history (two frames) and more history may be needed to contextualize itself. For the box cleanup task, the robot would often make an unsuccessful grasp attempt on the screwdriver. This is to be expected since it requires a precise grasp pose and there was a lot of variation in its initial placement across environment resets.

Table 13: Subgoal-level success rate (%), average progress reward (%), and overall task completion rate (%) across four real-world tasks.

| | Subgoal1 | Subgoal2 | Subgoal3 | Subgoal4 | Avg. Progress Reward | Overall Completion |
|---|---|---|---|---|---|---|
| **Banana Picking** | | | | | | |
| DiffPo (Pre) | 55 | 50 | N/A | N/A | 53 | 50 |
| RAVEN-abs | 95 | 90 | N/A | N/A | 93 | 90 |
| RAVEN | 100 | 100 | N/A | N/A | 100 | 100 |
| **Beans Scooping** | | | | | | |
| DiffPo (Pre) | 85 | 15 | 0 | 20 | 30 | 0 |
| RAVEN-abs | 100 | 85 | 65 | 80 | 83 | 55 |
| RAVEN | 100 | 75 | 50 | 55 | 70 | 45 |
| **Coffee Cleanup** | | | | | | |
| DiffPo (Pre) | 100 | 90 | 20 | 15 | 56 | 15 |
| RAVEN-abs | 100 | 95 | 95 | 80 | 93 | 80 |
| RAVEN | 75 | 75 | 80 | 70 | 75 | 45 |
| **Box Cleanup** | | | | | | |
| DiffPo (Pre) | 65 | 35 | 30 | N/A | 43 | 30 |
| RAVEN-abs | 100 | 65 | 65 | N/A | 77 | 65 |
| RAVEN | 100 | 70 | 60 | N/A | 77 | 60 |

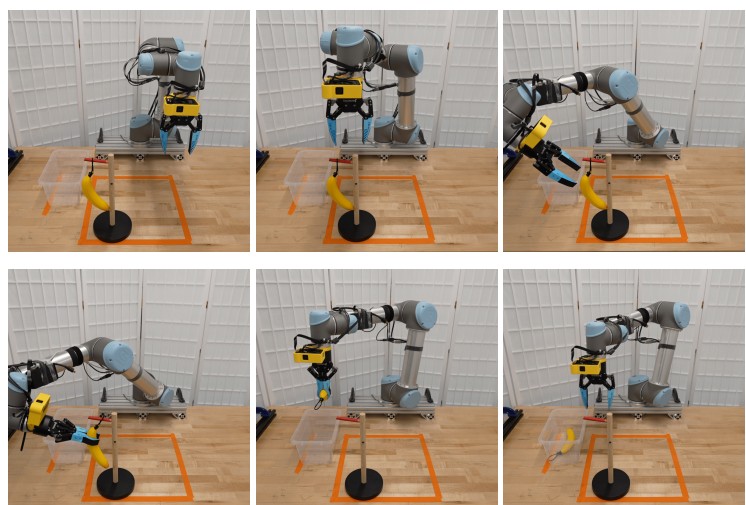

(a) Banana Picking

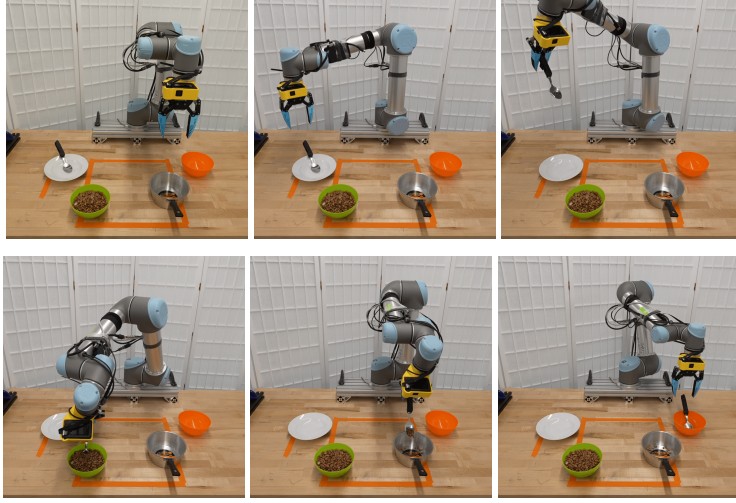

(b) Beans Scooping

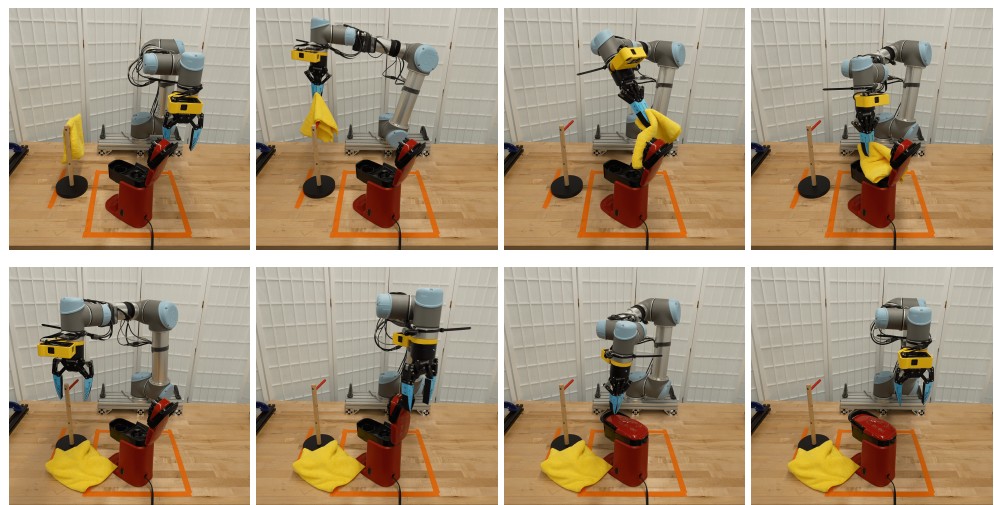

(c) Coffee Cleanup

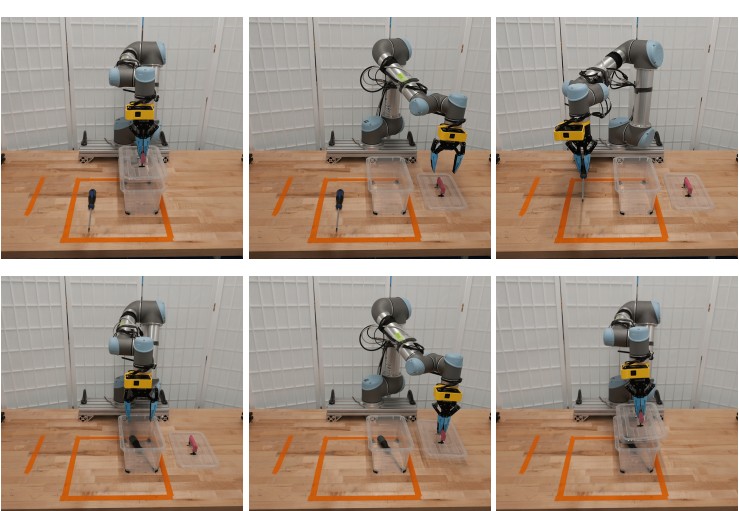

(d) Box Cleanup

Figure 12: Visualization of one episode for (a) Banana Picking, (b) Beans Scooping, (c) Coffee Cleanup, and (d) Box Cleanup. Each subfigure illustrates the trajectory of key action steps.

