# OpenReview forum: "RAVEN: End-to-end Equivariant Robot Learning with RGB Cameras"
_ICLR.cc/2026/Conference — ICLR 2026 Poster_

### Official Review · Reviewer_KKzi · 2025-10-24

**Soundness:** 3
**Presentation:** 3
**Contribution:** 3
**Rating:** 6
**Confidence:** 5

**Summary:**

This paper proposes an end-to-end approach for learning SE(3)-equivariant visuomotor policies directly from RGB input. The key idea is to convert image patches into geometric tokens. Each token carries canonicalized features in a local frame and an explicit SE(3) pose derived from the pixel ray geometry, using camera intrinsics/extrinsics. They leverage geometric transform attention (GTA) layer then computes attention using relative token poses, enabling SE(3)-equivariant attention. Finally, the decoder predicts end-effector (EEF)-relative actions via action tokens anchored at the current EEF pose. Results are reported on standard simulation benchmarks and real-robot tasks.
The approach is logical and practically appealing. While the experimental results over baselines are sometimes limited, the paper presents a clean pathway to SE(3)-aware RGB policies. The claims would be strengthened by broader EquiDiffPo comparisons, parity with 3D-input methods.

**Strengths:**

-A clear and principled route to achieving SE(3)-equivariance without relying on 3D point-cloud or voxel inputs—enabled by ray-based tokenization and pose-aware attention for end-to-end equivariant policy learning from image-based observations.
 -A coherent system architecture that serves as a practical guideline for leveraging SE(3) symmetry in pixel-based visuomotor control tasks.

**Weaknesses:**

- Novelty granularity: The core contribution is the "integration" of known components (ResNet features, ray back-projection, pose-aware attention, and an EEF-relative action head). The encoder is mostly standard aside from attaching per-patch poses. The paper's strongest novelty is at the system level, rather than in the encoder/decoder design individually.
- Experimental strength: Improvements over an equivariant baseline (e.g., EquiDiffPo) are often modest, though the baseline leverages SO(2)-symmetry. Given the emphasis on "SE(3) equivariance from 2D", it would be more compelling to match the performance from the 3D-input methods, (using e.g., point-cloud/voxel) under identical protocols [1,2].

<Typos>
p.6, 4.2., 3rd paragraph: convert he action geometric tokens" -> "convert the action geometric tokens"

[1] Wang et al., Equivariant Diffusion Policy (2024).
[2] Zhu et al., SE(3)-Equivariant Diffusion Policy in Spherical Fourier Space (2025).

**Questions:**

- Include EquiDiffPo comprehensively (at least all simulation tasks) for a fair comparison.
- Add comparisons to 3D input policies (point-cloud/voxel) to substantiate the benefits of RGB-only SE(3)-equivariance.

---

> ### Author Response · Authors · 2025-11-30
>
> [Part 1 of 2] We thank the reviewer for acknowledging that our work provides a clear and principled route to achieving SE(3)-equivariance directly from images, as well as a coherent architecture for leveraging symmetry in pixel-based visuomotor control. To address the concerns regarding novelty granularity, we added an additional ablation study in the main text to isolate the contribution of each component. To highlight the strength of our method compared to EquiDiff, we have added it as a baseline for the DexMimicGen experiment. Moreover, to respond to the request for comparisons with 3D-input policies, we aggregate results from works on 3D-based equivariant methods, demonstrating that our approach achieves relatively high success rates despite working on RGB-only inputs.
>
> >Novelty granularity: The core contribution is the "integration" of known components (ResNet features, ray back-projection, pose-aware attention, and an EEF-relative action head). The encoder is mostly standard aside from attaching per-patch poses. The paper's strongest novelty is at the system level, rather than in the encoder/decoder design individually.
>
> We would like to gently push back against this critique. Although our system integrates several known components, multiple aspects of our approach are novel beyond simply combining existing ideas. First, while some works have looked at action prediction in the end-effector frame (such as [1, 2]), ours is the first method that enables action prediction in the local frame of multiple end-effectors at once (e.g., bimanual setup). Our novel action decoder network is specifically designed for this purpose and is shown to empirically outperform the absolute action decoder networks (Conditional UNet with Vector Neurons) in Tables 8 and 9. Second, we propose a novel SE(3) ray encoding approach, which differs from the more standard practice of encoding rays in ray space. We show in the newly added ablation study (see end of Section 5.1) that this SE(3) ray encoding provides an average 4% boost in success rate on a subset of MimicGen tasks compared to the previous ray encoding used in GTA[3]. Lastly, our method uses a modified version of GTA that combines dot-product attention with negative Euclidean distance attention. This approach balances the stability of dot-product attention with the equivariance guarantees of distance-based attention, and to our knowledge has not appeared in prior work.
>
>
> >Experimental strength: Improvements over an equivariant baseline (e.g., EquiDiffPo) are often modest, though the baseline leverages SO(2)-symmetry.
>
> Yes, our method achieves an average 12% increase in success rate over EquiDiffPo on the MimicGen tasks. We appreciate the reviewer’s observation that one might expect an even larger performance gap given that RAVEN’s SE(3) equivariance should afford substantial generalization capabilities compared to EquiDiffPo’s SO(2) equivariance. However, MimicGen includes primarily tabletop tasks, so we do not expect to see a substantial benefit from our model’s ability to generalize to out-of-plane rotations (e.g. the tabletop being inverted).
>
> Even in this restricted tabletop setting, we still expect RAVEN to outperform EquiDiffPo. This is because EquiDiffPo treats the wrist-mounted camera as invariant (e.g. unchanged by transformations), whereas our model accurately models the SE(3) transformation action on wrist-mounted camera images. Therefore, RAVEN will generalize better even in the restricted SO(2) case. To show this empirically, we have added a comparison to EquiDiffPo in the DexMimicGen experiments (see results below or Table 2 in the updated paper). We find that RAVEN outperforms EquiDiffPo by an average of 20%. We hypothesize that the performance gap widens in the bimanual setting since there are two wrist-mounted cameras whose transformation properties EquiDiffPo cannot adequately capture.
>
> |  | Average | Threading | 3P Assembly | Transport | B. Cleanup | D. Cleanup | Tray Lift |
> | :-------         | :------: | :-------: |  :-------: |  :-------: | :------: | :-------: |  :-------: |
> | RAVEN       | 82 | 63 | 69 | 76 | 98 | 97 | 90 |
> | EquiDiffPo  | 62    |  29 | 45 | 63 | 85 | 94 | 56  |
>
> These new results also address the reviewer’s request to include EquiDiffPo comprehensively (at least all simulation tasks) for a fair comparison.

---

> > ### Author Response · Authors · 2025-11-30
> >
> > [Part 2 of 2]
> > >Given the emphasis on "SE(3) equivariance from 2D", it would be more compelling to match the performance from the 3D-input methods, (using e.g., point-cloud/voxel) under identical protocols
> >
> > We agree that it would be compelling to match the performance of methods that receive 3D spatial data. However, this work focuses on the narrow, yet important, problem of robot learning with RGB-only cameras. As evidenced by the popularity of robot frameworks/platforms like UMI and ALOHA, there is a demonstrated need to solve this problem without access to 3D inputs.
> >
> > To understand the current gap between our methods and 3D-input methods, we have compiled some numbers reported by [4] in the table below. This table shows the average success rate across all 12 MimicGen tasks. It is important to emphasize that these 3D methods leverage significantly richer observations: they fuse point clouds obtained from four agentview RGB-D cameras plus one in-hand RGB camera, yielding a near-complete reconstruction of the scene. In contrast, RAVEN operates with only one agentview RGB camera and one in-hand RGB camera. Despite this substantial difference in input richness, our method achieves competitive performance across MimicGen tasks (substantially outperforming DP3 and on par with EquiDiffpo (voxel)). While there remains a performance gap with EquiDiffPo (Point Cloud), it would be interesting to see how or if this gap could be closed using only RGB inputs in future work.
> >
> > |                | Average SR (%) |
> > | :-------         | :------: |
> > | RAVEN (RGB images)     | 66  |
> > | EquiDiffPo (Point Cloud)  | 77  |
> > | EquiDiffPo (Voxel)            | 64  |
> > | DP3 (Point Cloud)            | 24  |
> >
> >
> > [1] Chi, Cheng, et al. "Diffusion policy: Visuomotor policy learning via action diffusion." The International Journal of Robotics Research 44.10-11 (2025): 1684-1704.
> >
> > [2] Wang, Dian, et al. "A Practical Guide for Incorporating Symmetry in Diffusion Policy." arXiv preprint arXiv:2505.13431 (2025).
> >
> > [3] Miyato, Takeru, et al. "Gta: A geometry-aware attention mechanism for multi-view transformers." arXiv preprint arXiv:2310.10375 (2023).
> >
> > [4] Wang, Dian. Equivariant Policy Learning for Robotic Manipulation. Diss. Northeastern University, 2025.

---

### Official Review · Reviewer_q8H3 · 2025-10-30

**Soundness:** 2
**Presentation:** 3
**Contribution:** 2
**Rating:** 2
**Confidence:** 4

**Summary:**

This paper presents RAVEN, the first SE(3)-equivariant policy learning framework that operates directly on RGB image observations. The key innovation is treating image patches as collections of 3D rays that transform under SE(3) roto-translations, enabling equivariant processing through geometric tokens.

**Strengths:**

- The ray-based formulation for achieving SE(3) equivariance from RGB images is elegant and principled. Converting image patches to geometric tokens with SE(3) poses is a creative solution to a real problem in robotics.
- The experiments cover a range of test settings, and the ablation studies help understand each component of the method.
- Overall the experiment results are good, which demonstrates the effectiveness of the proposed method.

**Weaknesses:**

- **Limited Theoretical Analysis of Ray-Patch Representation**: While the paper states that "each grid cell in the feature map is associated with an element of SE(3)", the justification for how a patch of pixels/rays defines a unique orientation is somewhat hand-waved (Figure 2). A more rigorous analysis of when this representation is well-defined and how discretization affects equivariance would strengthen the work.

- **Incomplete Equivariance Analysis**: The viewpoint generalization task (Section 5.1) perturbs only a single camera, which "breaks global equivariance" as the authors note. This seems to contradict the claimed advantage of camera-agnostic operation ("remaining agnostic to the number and placement of cameras," line 48).

- **Real-World Experiments Limited in Scope**:
  - Only 4 tasks with relatively simple manipulation primitives are considered.
  - No analysis of failure modes or edge cases is provided.

**Questions:**

- **Ray Discretization**: How does the grid resolution (5×5 in your implementation) affect the quality of the ray-based representation and overall performance?

- **Viewpoint Generalization Paradox**: If the method is truly camera-agnostic, why does performance drop when only the agent-view camera is perturbed (Table 3)? Doesn't this suggest the equivariance guarantee is weaker than claimed?

- **Computational Cost**: What is the actual inference time compared to baselines? Training time is reported, but deployment efficiency matters for robotics.

- **Flow Matching vs Diffusion**: Is the performance gain primarily from the ray-based encoder or from using flow matching instead of diffusion? An ablation would clarify this.

- **Comparison with [1]**: This work also claims RGB-based SO(3) equivariance. What are the key differences, and why wasn't this included as a baseline?


[1] Boce Hu, Xupeng Zhu, Dian Wang, Zihao Dong, Haojie Huang, Chenghao Wang, Robin Walters, and Robert Platt. Orbitgrasp: Se(3)-equivariant grasp learning. arXiv preprint arXiv:2407.03531

---

> ### Author Response · Authors · 2025-11-30
>
> [Part 1 of 4] We thank the reviewer for the feedback and for recognizing our method as elegant and principled. To address the main concern regarding the limited theoretical analysis of the ray-patch representation, we have added a new section in the main paper that provides a detailed analysis and clarifies our definition of equivariance. In addition, we include a real-world failure mode analysis and several new experiments that directly respond to the reviewer’s questions. Please see the detailed responses below.
>
> > Limited Theoretical Analysis of Ray-Patch Representation: While the paper states that "each grid cell in the feature map is associated with an element of SE(3)", the justification for how a patch of pixels/rays defines a unique orientation is somewhat hand-waved (Figure 2). A more rigorous analysis of when this representation is well-defined and how discretization affects equivariance would strengthen the work.
>
> We have updated that part of the method section to improve clarity. We will try to explain the idea here with an extreme example. Consider the case that we want to represent an entire image as a single ray feature. If we model this feature as  an element of ray space (R^3 x S^2), then we are implicitly saying that 2D rotations of the image should leave the geometric representation unchanged (the representation would be 2D rotation invariant). However, we know that natural images (and image patches) have an implicit “up” direction that varies under 2D rotations, meaning that they are 2D rotation equivariant. In order to capture the equivariance of the image patch to these 2D rotations, we associate each patch with an element of SE(3). In other words, we inject an explicit “up” direction of the image into the ray-space. It is important to note that this design choice is not necessary to achieve SE(3) equivariance, but it provides the network with more useful geometric bias to generate 3D understanding from image inputs.
>
> We also empirically justify the decision to encode rays as SE(3) features in a newly added ablation study (see Table 4 in the updated paper). In this ablation, we replace our SE(3) ray encoding with a simpler encoding where each ray is represented only by the patch’s center pixel direction (i.e., an S^2 direction vector rather than an SO(3) orientation). The results are reproduced in the table below, and reported as success rate (%) on MimicGen tasks across three seeds. We find an average drop in success rate of 4%, which supports the benefits of representing ray patches as elements of SE(3).
>
> |  | Stack D1 | Stack Three D1 | Square D2 | Hammer Cleanup D1 |
> | :-------      | :------: | :-------: |  :-------: |  :-------: |
> | RAVEN w/ Flow Matching   | 100    |  81    |    50  |    73   |
> |  RAVEN w/ Diffusion           | 100    |  74    |    44  |    69   |
>
>
> >Incomplete Equivariance Analysis: The viewpoint generalization task (Section 5.1) perturbs only a single camera, which "breaks global equivariance" as the authors note. This seems to contradict the claimed advantage of camera-agnostic operation ("remaining agnostic to the number and placement of cameras," line 48).
>
> The equivariance defined in RAVEN is with respect to global SE(3) transformations of the entire environment, including the objects, robot, and all sensors). Under this definition, perturbing only one single camera indeed breaks global equivariance. The claimed advantage of being “agnostic” refers to the robot platform’s camera configuration on which the model is trained, not to arbitrary per-camera perturbations at test time. Prior works on image-based equivariant policy learning are designed for a particular number (one wrist-mounted camera in [1]) or placement (top-down or nearly top-down in [2]) of RGB cameras. In contrast, our method can operate with any number of cameras (one or many) and any placement without architectural changes,, which greatly expands its applicability. We have updated the writing in the introduction to make this point more clear and avoid the previous ambiguity that may have led to misunderstanding.

---

> > ### Author Response · Authors · 2025-11-30
> >
> > [Part 2 of 4]
> > >Real-World Experiments Limited in Scope: Only 4 tasks with relatively simple manipulation primitives are considered. No analysis of failure modes or edge cases is provided.
> >
> > We would like to gently push back against the comment that our real-world tasks involved “relatively simple manipulation primitives”. The tasks included picking, placing, scooping and wiping. The difficulty of the tasks (especially in such a limited data regime of 75 or fewer demos) is evidenced by the low performance of DiffPo, the SOTA method, on these tasks in the same settings. Our real-world results therefore already demonstrate that RAVEN achieves significantly higher success rates than DiffPo, indicating that the method is effective even on challenging, contact-rich tasks. If helpful, the reviewer may also refer to the videos in the supplementary material, which show full task executions and highlight the complexity.
> > We have also added a discussion of common failure modes to Appendix H in the updated paper. Most failure modes of RAVEN are imprecise motions (e.g., failing to grasp the screwdriver handle) or incorrect subtask sequencing (e.g., getting stuck repeatedly pouring beans without moving on). These failure modes seem to be common in the baselines as well, and we did not observe any behaviors that could be directly associated with the particular equivariance properties or architecture design of our method.
> >
> > > Ray Discretization: How does the grid resolution (5×5 in your implementation) affect the quality of the ray-based representation and overall performance?
> >
> > The grid resolution does not affect the SE(3) equivariance of our method, but does impact performance. The SE(3) equivariance of our method is based on SE(3) actions applied to the camera poses and the scene contents, while the images themselves remain invariant under this global transformation (see Equation 6 in the paper).  So, although downsampling in the encoder may reduce equivariance with respect to image shifts, it will not affect equivariance to the global SE(3) transformations that we study here.
> > Importantly, the grid resolution is not manually chosen. In simulation, the 5×5 grid arises naturally from feeding the 76×76 camera images through the pretrained ResNet encoder. In real-world experiments, the 224×224 inputs similarly produce a 7×7 feature map. We have updated the paper to clarify this point.
> > To understand the effect on performance, we added an experiment that compares performance at three different grid resolutions: 3x3, 5x5 (default) and 7x7 in simulation. To achieve each grid resolution, we modified the downsampling in the pretrained ResNet encoder; the rest of the model and training procedure were held constant. The results are shown in the table below, averaged across three seeds. We find that 5x5 and 7x7 grid resolutions perform comparably, while 3x3 has a noticeable drop in performance.
> > We hypothesize that for high-precision manipulation tasks, even higher grid resolutions may be beneficial. However, attention scales quadratically with the number of tokens, so there is a tradeoff.
> >
> > |  | Stack D1 | Stack Three D1 | Square D2 | Hammer Cleanup D1 |
> > | :-------      | :------: | :-------: |  :-------: |  :-------: |
> > | 3x3 grid   |   99    |  73    |    37  |    67   |
> > | 5x5 grid   | 100    |  81    |    50  |    73   |
> > | 7x7 grid   |  100   | 79     |    47  |   74    |
> >
> > We have included this experiment in the Appendix of the updated paper.
> >
> > >Viewpoint Generalization Paradox: If the method is truly camera-agnostic, why does performance drop when only the agent-view camera is perturbed (Table 3)? Doesn't this suggest the equivariance guarantee is weaker than claimed?
> >
> > This is a good question. The equivariance guarantee in RAVEN is defined with respect to global SE(3) transformations of the entire environment (including objects, robots, and all sensors). So the drop in performance in Table 3 is expected, since the SE(3) transformation is applied to a single camera, not the entire environment, and thus does not fall under the equivariance guarantee.  Even so, RAVEN still maintains performance better than baselines in this experiment because it captures the 3D geometric structure contained in an image.
> >
> > We have received similar comments from other reviewers about our method’s equivariance guarantees, and want to avoid ambiguity as well as ensure that the claims are clear in the paper. In the updated version of the paper, we updated the writing to clarify that the equivariance is with respect to global transformations of the environment. In addition, we expanded Section 4.3 to explain the equivariance properties of our method and to contextualize them with respect to prior equivariant robot learning works.

---

> > > ### Author Response · Authors · 2025-11-30
> > >
> > > [Part 3 of 4]
> > > >Computational Cost: What is the actual inference time compared to baselines? Training time is reported, but deployment efficiency matters for robotics.
> > >
> > > The inference times are 55 ms for RAVEN, 70 ms for DiffPo and 85 ms for EquiDiffPo. We timed inference with the two-camera observation setting using the same hardware setup as used to report training times in the paper. RAVEN is slightly faster than EquiDiffPo and DiffPo because its transformer decoder is more lightweight than their UNet decoder. Also, RAVEN uses 10 flow-matching denoising steps whereas EquiDiffPo and DiffPo require 16 denoising steps (with DDIM). We agree that deployment efficiency is key in robotics, since high frequency control is necessary for smooth motions in contact-rich tasks.  A lot of the gains in inference speed will come from innovations with denoising schemes, which was not the focus of our paper.
> > >
> > > We believe training time also plays a key role in the usability of imitation learning methods.  Demonstrations can be costly or difficult to collect, and it is often not clear how many demonstrations are needed until after the model is trained and evaluated. The faster a model trains, the faster the iteration cycles, which allows users to more quickly decide whether to collect more demonstrations or to adjust the demonstrated behavior.
> > >
> > > >Flow Matching vs Diffusion: Is the performance gain primarily from the ray-based encoder or from using flow matching instead of diffusion? An ablation would clarify this.
> > >
> > > That is a good question. During development, we explored both flow matching and diffusion for RAVEN.  We found that flow-matching provides a small performance boost on some, but not all, tasks. We will include a head-to-head comparison in the Appendix of the updated paper. Results on a subset of MimicGen tasks are shown in the table below (averaged over three seeds). It is worth noting that we did not extensively tune either the flow matching or diffusion hyperparameters, and directly adopted the hyperparameters from existing works (PointFlowMatch[3] for flow matching and EquiDiffPo[2] for diffusion).
> > >
> > > |  | Stack D1 | Stack Three D1 | Square D2 | Hammer Cleanup D1 |
> > > | :-------      | :------: | :-------: |  :-------: |  :-------: |
> > > | RAVEN w/ Flow Matching   | 100    |  81    |    50  |    73   |
> > > |  RAVEN w/ Diffusion           | 100    |  84    |    43  |    70   |

---

> > > > ### Author Response · Authors · 2025-11-30
> > > >
> > > > [Part 4 of 4]
> > > > > This work also claims RGB-based SO(3) equivariance. What are the key differences, and why wasn't this included as a baseline?
> > > >
> > > > The linked paper cited by the reviewer, “Orbitgrasp”, in fact focuses on a different setting: robotic grasping with point cloud inputs. We therefore believe the reviewer is referring to another work from the same authors, ISP [1], which is indeed closely related to ours.
> > > > There are two key differences between ISP and RAVEN. First, ISP is designed specifically for observations from a single wrist-mounted camera whereas RAVEN can handle observations from any number of cameras, either static or moving.  This means RAVEN can be used with significantly more robotic platforms and can solve tasks where the wrist-mounted camera view is insufficient or suffers from occlusion. The other key difference is in the way that equivariance is achieved. ISP achieves SO(3) equivariance by mapping image features to irreducible representations of the sphere, and decodes actions using a conditional UNet. In contrast, RAVEN achieves SE(3) equivariance by mapping image features to SE(3) ray features, processing and decoding actions with GTA layers.  We did not include ISP as a baseline in the paper because it is tailored for the single, wrist-mounted camera setting, whereas we looked at the multi-camera setting.
> > > >
> > > > Nevertheless, for completeness, we conducted a small empirical comparison of the two methods on the threading and three-piece assembly tasks from MimicGen, following the same experimental setup used in the main table. These tasks were selected to highlight the challenge of using a single wrist-mounted camera: when the robot goes to manipulate one object, the camera can lose view of other relevant information in the scene.  The comparison is conducted under different task difficulty levels: D0 and D2. As shown in the table below, both RAVEN and ISP perform well under the D0 setting, where object placement randomness is minimal and the task largely reduces to memorizing goal locations. Even in this simple regime, RAVEN outperforms ISP by 10%. However, under the D2 setting, both methods exhibit a substantial performance drop. The main reason is that the wrist-mounted camera view becomes heavily occluded by the grasped object, resulting in insufficient visual information. In this scenario, the model can no longer rely on memorizing the target position. Despite this drop, RAVEN remains 15% better than ISP, primarily because it naturally integrates information from multiple camera views, which supports it to recover from occlusions and maintain reliable goal perception. Since ISP is explicitly designed for single-view input, it cannot easily handle tasks that require multi-view observations or be directly adapted to new camera configurations. In contrast, RAVEN, by design, can operate with any number of views without architectural changes.
> > > >
> > > > |   | 3P Assembly D0 | 3P Assembly D2 | Threading D0 | Threading D2 |
> > > > | :-------         | :------: | :-------: |  :-------: |  :-------: |
> > > > | RAVEN       | 82   | 27 | 95 | 28 |
> > > > | ISP [1]         | 75  |  11 | 85 | 14 |
> > > >
> > > >
> > > > [1] Hu, Boce, et al. "3D Equivariant Visuomotor Policy Learning via Spherical Projection." arXiv preprint arXiv:2505.16969 (2025).
> > > >
> > > > [2] Wang, Dian, et al. "Equivariant Diffusion Policy." Conference on Robot Learning. PMLR, 2025.
> > > >
> > > > [3] Chisari, Eugenio, et al. "Learning robotic manipulation policies from point clouds with conditional flow matching." arXiv preprint arXiv:2409.07343 (2024).

---

### Official Review · Reviewer_DW7R · 2025-11-01

**Soundness:** 3
**Presentation:** 3
**Contribution:** 3
**Rating:** 6
**Confidence:** 5

**Summary:**

The paper proposes a ray-based SE(3) equivariant visual manipulation policy that can be used end-to-end from RGB images without involving traditonal 3D modalities like point clouds. Speficially, a feature map from pretrained 2D image model is lifted to featured rays using known camera parameters. These featured rays are grouped to form a SE(3)-posed geometric token. A series of transformer blocks with Geometric transform attention (GTA) is then applied to equivariantly process visuo-proprioceptive information. Finally, these equivariant representations are decoded into flow-matching prediction. Experimental results on standard benchmarks confirm clear advantage of the proposed method.

**Strengths:**

### [Strength 1] Novelty
As far as I know, this is the first ray-based equivariant robot policy. While [1] has previously explored equivariant models in Plucker space by leveraging its homogeneous space property as being SE(3)/(SO(2)xR), it was limited to computer-vision oriented applications. As ray-based representation is becoming increasingly popular in 3D vision fields [2,3,4], it is timely to explore its application to robotics field.

### [Strength 2] Scalability
Most existing SE(3) equivariant manipulation methods are not scalable as they rely on complicated and model-specific operations that are not well-optimized. On the other hand. the proposed method is more compatible with more commonly used operations that are heavily optimized. For instance, GTA is compatible with flash attention, which can greatly improve memory efficiency and training speed.

[1] Xu et al. ""Equivariant Light Field Convolution and Transformer."

[2] Jin et al. "Lvsm: A large view synthesis model with minimal 3d inductive bias."

[3] Jiang et al. "RayZer: A Self-supervised Large View Synthesis Model."

[4] Li et al. "Cameras as relative positional encoding."

**Weaknesses:**

### [Weakness 1] Definition of Equivariance is Questionable
The claimed SE(3) equivariance property of the proposed method is defined with respective to the gripper and camera poses. This is a valid and helpful inductive bias especially when denoising multiple gripper pose trajectory or incorporating multiview information. However, it should be noted that this kind of definition is different from more accepted defintion of equivariant policy where the equivariance of the action is defined with respect to the content of the scene, not to the camera. Assuming correct point cloud registration, scene-to-gripper equivariance is a strict superset of this camera-to-gripper equivariance. As such, it is not quite true that this is "first end-to-end equivariant policy learning method that works with image-based observations." In fact, this is a shared limitation of other equivariant methods based on ray [1] or spherical projection [2].

[1] Xu et al., "Equivariant Light Field Convolution and Transformer."

[2] Hu et al., "3D Equivariant Visuomotor Policy Learning via Spherical Projection"

### [Weakness 2] Effectiveness of Equivariance is Questionable
Experimental results are solid enough to support the effectiveness of the proposed method. However, it is unclear equivariance is indeed the main factor of this observed advantage. The use of GTA-like camera RoPE approach has been reported to be effective in several non-equivariant methods [3,4,5]. Hence, unless an ablation without equivariance is provided (i.e., using nonequivariant MLP), I would be inclined to believe that the performance advantage is mainly from this relative attention property, not from equivariance. In terms of benchmark, I think it is more fair to compare the baseline diffusion policy with the same GTA architecture (but with non-equivariant counterpart) than using the original Unet-based architecture.

Also, it is unclear why ray-based representation is necessary to achieve the equivariance. Unlike [1] which used homogeneous space convolution, the proposed method lifts rays onto SE(3) space without ever being grounded to ray space again. If so, why not simply tokenize a featured point cloud projected to a unit sphere in the local camera reference frame, and then lifting it onto SE(3)?

[3] Miyato et al. "Gta: A geometry-aware attention mechanism for multi-view transformers."

[4] Kong et al. "Eschernet: A generative model for scalable view synthesis."

[5] Li et al. "Cameras as relative positional encoding."

**Questions:**

Please find my questions in the weaknesses section.

---

> ### Author Response · Authors · 2025-11-30
>
> [Part 1 of 2]
> We thank the reviewer for the constructive feedback. We appreciate the recognition of our method’s novelty and scalability, particularly its use of simple, well-optimized operations compared to prior SE(3)-equivariant approaches. To address Weakness 1, we have added a complete equivariance analysis in a new section of the main paper to clarify our formulation. To address Weakness 2, we include an ablation study demonstrating the contribution of equivariance to the final performance improvements. Detailed responses to each point are provided below.
>
> > The claimed SE(3) equivariance property of the proposed method is defined with respective to the gripper and camera poses. … However, it should be noted that this kind of definition is different from more accepted defintion of equivariant policy where the equivariance of the action is defined with respect to the content of the scene, not to the camera.
>
> We think the reviewer makes a great point. We agree that the claim of SE(3) equivariance in robot learning literature generally refers to transformation actions on the robot and objects but not the sensors (although we will show how this definition is sometimes used loosely below). We also agree that this equivariance is a superset of the SE(3) equivariance of our method. We updated the writing to clearly state this distinction. We also find the idea of distinguishing between “scene-to-gripper” and “camera-to-gripper” equivariance very helpful.
>
> To say more about the nuance of this distinction between equivariance claims, it is worth exploring the claims in some of the prior works. In [1], the observation is a camera image with a top-down view. So, rotating the image corresponds to rotating the objects and the robot, separate from the camera. Since it is easy to define a mapping that rotates an image, we can guarantee that a 2D rotationally-equivariant network will be generalized to rotations of the objects and the robot. In works like [2, 3], the input is a point cloud generated by several depth sensors. In the perfect world, transforming a point cloud corresponds to transforming the objects and the robot, separate from the sensors. However, in reality, unless enough sensors are used to eliminate all occlusion, transforming the point cloud corresponds to transforming the objects, robot and sensors (the same equivariance guarantee as our method!). Empirically, the point cloud-based equivariant methods have strong generalization capabilities, suggesting that the impact of occlusion regions is small.
>
>
> > Experimental results are solid enough to support the effectiveness of the proposed method. However, it is unclear equivariance is indeed the main factor of this observed advantage. The use of GTA-like camera RoPE approach has been reported to be effective in several non-equivariant methods [3,4,5]. Hence, unless an ablation without equivariance is provided (i.e., using nonequivariant MLP), I would be inclined to believe that the performance advantage is mainly from this relative attention property, not from equivariance. In terms of benchmark, I think it is more fair to compare the baseline diffusion policy with the same GTA architecture (but with non-equivariant counterpart) than using the original Unet-based architecture.
>
> For benchmarks, we intentionally chose to use the architectures exactly as proposed in the original papers,  since evaluating absolute performance requires comparing against the baselines in their original, unmodified form rather than introducing structural changes that could create unfair or suboptimal configurations. However, we agree that such a comparison against non-equivariant variants is useful for understanding the proposed work. Accordingly, we have added an ablation study that includes this comparison (see Table 4 of the updated paper).  Specifically, we look at an ablation of RAVEN that retains the same encoder and decoder architecture but with non-equivariant layers (we replaced GTA with standard dot-product attention and absolute position embeddings).  We reproduce the results in the table below (MimicGen tasks; averaged across 3 seeds).
> Removing the equivariant layers in RAVEN results in between a 10 and 25% drop in success rate.  Note that the ablated model still received the camera intrinsics and extrinsics for each image as input, so this experiment highlights the impact of equivariance on RAVEN’s performance.
>
> |  | Stack D1 | Stack Three D1 | Square D2 | Hammer Cleanup D1 |
> | :-------      | :------: | :-------: |  :-------: |  :-------: |
> | RAVEN                                                        | 100    |  81    |    50  |    73   |
> | RAVEN w/ Non-Equi Encoder & Decoder   |  90     | 56     | 27      | 60    |

---

> > ### Author Response · Authors · 2025-11-30
> >
> > [Part 2 of 2]
> > > Also, it is unclear why ray-based representation is necessary to achieve the equivariance. Unlike [1] which used homogeneous space convolution, the proposed method lifts rays onto SE(3) space without ever being grounded to ray space again. If so, why not simply tokenize a featured point cloud projected to a unit sphere in the local camera reference frame, and then lifting it onto SE(3)?
> >
> > The ray-based representation we propose is not necessary to achieve SE(3) equivariance.  Our design was influenced by the need to be computationally efficient (to allow real-time control with multiple image observations) and general-purpose (a single architecture that performs well across different tasks). While the approach from [4] could also be used to achieve SE(3) equivariance, it has some notable limitations for robot learning in practical settings.  First, its ray-to-ray convolution relies on defining local ray neighborhoods, but it is unclear how to set these neighborhoods consistently across different camera intrinsics or camera configurations. Second, it proposes a ray-to-point convolution to generate representations over 3D points, but it is not obvious how to select such points in a task-agnostic manner.
> >
> > We do not fully understand the reviewer’s final suggestion, since we do not have access to point clouds in our setting, so tokenizing a point cloud projected to a unit sphere is not directly feasible. There are some works [5, 6] that lift image features onto the sphere in the local frame but we are not aware of any that lift to SE(3).  While it may be worth exploring how best to do this, we want to note that representing features on the sphere often requires higher-order irreps of SO(3) which can be computationally inefficient to process.
> >
> > Finally, we want to call the reviewer’s attention to some new experimental results we added to the paper. First, in an ablation study (see Section 5.1), we show that RAVEN’s performance drops 4% when we modify the proposed SE(3) ray feature encoding to use an R^3 x S^2 ray feature encoding (as used in GTA [7]), which demonstrates the benefit of our encoding scheme. Second, in the Appendix, we show that reducing the number of ray tokens per image also reduces performance, indicating that the expressiveness of the ray representation DOES matter.
> >
> > [1] Wang, Dian, Robin Walters, and Robert Platt. "$\mathrm {SO}(2) $-Equivariant Reinforcement Learning." arXiv preprint arXiv:2203.04439 (2022).
> >
> > [2] Yang, Jingyun, et al. "Equibot: Sim (3)-equivariant diffusion policy for generalizable and data efficient learning." arXiv preprint arXiv:2407.01479 (2024).
> >
> > [3] Ryu, Hyunwoo, et al. "Diffusion-edfs: Bi-equivariant denoising generative modeling on se (3) for visual robotic manipulation." Proceedings of the IEEE/CVF Conference on Computer Vision and Pattern Recognition. 2024.
> >
> > [4] Xu, Yinshuang, Jiahui Lei, and Kostas Daniilidis. "Equivariant Light Field Convolution and Transformer." arXiv preprint arXiv:2212.14871 (2022).
> >
> > [5] Klee, David M., et al. "Image to sphere: Learning equivariant features for efficient pose prediction." arXiv preprint arXiv:2302.13926 (2023).
> >
> > [6] Hu, Boce, et al. "3D Equivariant Visuomotor Policy Learning via Spherical Projection." arXiv preprint arXiv:2505.16969 (2025).
> >
> > [7] Miyato, Takeru, et al. "Gta: A geometry-aware attention mechanism for multi-view transformers." arXiv preprint arXiv:2310.10375 (2023).

---

### Official Review · Reviewer_rWp8 · 2025-11-03

**Soundness:** 3
**Presentation:** 2
**Contribution:** 2
**Rating:** 4
**Confidence:** 3

**Summary:**

This paper presents RAVEN, an end-to-end SE(3)-equivariant policy learning framework that operates using only RGB camera inputs. The key idea is to interpret 2D image patches as 3D rays defined by known camera intrinsics and extrinsics, which allows the use of SE(3)-equivariant geometric transformers to process visual observations. The encoder outputs geometric tokens that can be fused across arbitrary camera views, and the policy is trained with flow matching to predict continuous-time action trajectories. Empirically, RAVEN achieves strong results on MimicGen and DexMimicGen benchmarks and shows faster training and better viewpoint generalization compared to prior equivariant or diffusion-based policies.

**Strengths:**

The work introduces a creative and elegant approach to extend SE(3)-equivariance to RGB-only observations, bridging a long-standing gap between equivariant robotic policies (which typically rely on structured 3D inputs) and practical camera-based systems. The ray-based tokenization formulation is novel and theoretically sound.

RAVEN emperically outperforms strong baselines across multiple tasks and even surpasses pretrained diffusion policies while training faster. The evaluation covers both simulation and real-world tasks.

**Weaknesses:**

Limited applicability in realistic deployments: The method’s core assumption — that camera extrinsics and intrinsics are precisely known and remain static — limits its applicability in unstructured real-world settings. The real-world experiments use carefully calibrated, fixed setups, so the paper does not convincingly demonstrate robustness to realistic calibration noise.

Missing ablations / comparisons on input modality and architecture choices: The evaluation focuses exclusively on RGB input. Despite claiming compatibility with multimodal data, the paper lacks any ablation or comparison using non-equivariant model + depth, RGB-D, or depth-estimation-based representations, which are standard in 3D-aware robot learning literature. It is therefore unclear whether the gains stem from true equivariance or from better geometric bias in the encoder.

**Questions:**

1. Camera calibration sensitivity: How robust is RAVEN to small errors in extrinsic or intrinsic calibration? Could the authors quantify the degradation in performance?

2. Depth and RGB-D ablation: Have the authors tried integrating depth maps or estimated depth (e.g., from pretrained monocular models) into the ray-based representation? This would make a strong case for the claimed modality-agnostic design.

3. Failure analysis: It would be useful to include qualitative failure cases — for instance, where equivariance assumptions break or where viewpoint perturbations cause prediction drift — to better understand the model’s limits.

---

> ### Author Response · Authors · 2025-11-30
>
> [Part 1 of 3] We thank the reviewer for the comments. We are glad to hear that the reviewer finds our work creative and elegant, and sees its potential to bridge a long-standing gap between equivariant robotic policies and practical camera-based systems. To address the reviewer’s concerns regarding applicability and camera sensitivity in realistic settings, we have added a thorough experiment that injects errors into the camera parameters during both training and inference. In addition, we included an ablation study to analyze the effects of different input modalities and architectural choices. Detailed responses to each point are provided below.
>
> > Limited applicability in realistic deployments: The method’s core assumption — that camera extrinsics and intrinsics are precisely known and remain static — limits its applicability in unstructured real-world settings. The real-world experiments use carefully calibrated, fixed setups, so the paper does not convincingly demonstrate robustness to realistic calibration noise.
>
> First, we would like to clarify that our method does not assume that camera extrinsics remain static.  All the experiments include wrist-mounted cameras where the camera extrinsic changes as the robot moves. We also explored the case where the table-mounted (i.e. agent-view) camera pose changes and our method outperforms all baselines by an average of 23% (see “Viewpoint Generalization” in the experiment section). Nonetheless, we understand the reviewer’s overall critique, as camera parameters may not be exact in the real world due to calibration error or variations between robots in a fleet. In the response below, we empirically test our method’s robustness to camera parameter variations and show that it still performs well with noisy calibration.
>
> Lastly, we want to stress that having accurately calibrated cameras is not tied to whether a setting is “unstructured.” In fact, most popular manipulation frameworks (e.g. UMI [1], ALOHA[2]) mount cameras rigidly on the gripper or robot base, meaning the extrinsics are known, even with small errors, regardless of the environment in which it is deployed.
>
> > Camera calibration sensitivity: How robust is RAVEN to small errors in extrinsic or intrinsic calibration? Could the authors quantify the degradation in performance?
>
> These are great questions. We have added a new experiment to evaluate the robustness of RAVEN to errors in the camera parameters, which is included in the updated paper. We look at two cases: (1) training with perfect calibration but testing with imperfect calibration, and (2) using imperfect calibration during both training and testing (which reflects the real-world scenario).
> For imperfect calibration, we inject a reasonable amount of noise by offsetting the simulator-provided ground-truth parameters. Specifically, we apply a  5-pixel (6.6%) shift to the principal point, a 2% variation in focal length, a 0.5 - 1.5 cm perturbation to the camera translation, and a 6, -4, 8 degree variation to the camera roll, pitch, and yaw angles. The results in the table below show that Case 1 suffers a larger drop in performance than Case 2. This makes sense since the model can learn to correct for imperfect calibration during training, but may struggle when such errors appear only at test time. We further looked at Case 1 but added data augmentation to the camera parameters during training. This greatly improved performance, achieving the same success rate as RAVEN with perfect calibration on three out of four tasks. This demonstrates that, even without knowing the exact calibration errors, camera-parameter augmentation can effectively compensate for them.
>
> |  | Stack D1 | Stack Three D1 | Square D2 | Hammer Cleanup D1 |
> | :-------      | :------: | :-------: |  :-------: |  :-------: |
> | RAVEN w/ perfect calibration       |   100    |  81    |    50  |    73   |
> | Case 1                      |   100    |  76    |    42  |    69   |
> | Case 2                      | 100    |  80    |    49  |    71   |
> | Case 1 with data augmentation  |  100   | 81     |    47  |   73    |

---

> > ### Author Response · Authors · 2025-11-30
> >
> > [Part 2 of 3]
> > >Depth and RGB-D ablation: Have the authors tried integrating depth maps or estimated depth (e.g., from pretrained monocular models) into the ray-based representation? This would make a strong case for the claimed modality-agnostic design.
> >
> > Our main focus with this work is on RGB cameras, which remain the dominant modality in large-scale datasets [3,4] and common hardware setups [1,2]. Nonetheless, we are interested in extending RAVEN to incorporate depth information. For this response, we created a simple variant of RAVEN that incorporates depth information from RGBD sensor by processing the depth image with a small convolutional network and concatenating it channel-wise with the ResNet encoder’s feature map. We evaluated this model on a subset of MimicGen tasks and report the success rates, averaged across 3 seeds, in the table below. Unexpectedly, we find that RAVEN with RGBD does not outperform RAVEN with RGB. Due to the time constraints of this rebuttal process, we were not able to iterate on the design here, but this does demonstrate that adding depth to the framework is feasible.
> > |  | Stack D1 | Stack Three D1 | Square D2 | Hammer Cleanup D1 |
> > | :-------      | :------: | :-------: |  :-------: |  :-------: |
> > | RAVEN w/ RGB  |   100    |  81    |    50  |    73   |
> > | RAVEN w/ RGBD   |   99    |  78    |    35  |    74   |
> >
> > While we believe RAVEN is well-suited to integrate other observation modalities (and list ways to implement it in the Appendix), we acknowledge that the current paper does not demonstrate such capability.  We have edited the wording in the first contribution statement to avoid any confusion.
> >
> > > Missing ablations / comparisons on input modality and architecture choices: The evaluation focuses exclusively on RGB input. Despite claiming compatibility with multimodal data, the paper lacks any ablation or comparison using non-equivariant model + depth, RGB-D, or depth-estimation-based representations, which are standard in 3D-aware robot learning literature. It is therefore unclear whether the gains stem from true equivariance or from better geometric bias in the encoder.
> >
> > We have added an expanded ablation study to test how different components in RAVEN contribute to performance. The results reported are success rates (%) on a subset of MimicGen tasks, shown in the table below (or Section 5.1 in the updated paper). In “RAVEN w/o SE(3) Ray Encoding”, we substitute the proposed ray encoding scheme, where the feature map elements have an up-direction, with the method used in GTA [5], where the ray direction is represented with embedding on pixel coordinates. Here, we see a small drop in performance (4 % averaged across tasks), which indicates the utility of a stronger geometric bias in our ray encoding. In “RAVEN w/o Equi Decoder” and “RAVEN w/o Equi Encoder & Equi Decoder”, we replace the GTA layers with standard dot-product attention in the decoder and in the full network, respectively. This removes the equivariance, while still preserving the geometric information (the tokens contain information about pose from absolute positional embeddings). The results show a 4% drop in performance when equivariance is removed from the decoder and an additional 14% drop when equivariance is removed from the encoder. This demonstrates that RAVEN’s equivariance also contributes to its performance when geometric information is held constant.
> >
> > |  | Stack D1 | Stack Three D1 | Square D2 | Hammer Cleanup D1 |
> > | :-------      | :------: | :-------: |  :-------: |  :-------: |
> > | RAVEN                                                             |   100    |  81    |    50  |    73   |
> > | RAVEN w/o SE(3) Ray features                        | 100    |  74    |    44  |    69   |
> > | RAVEN w/o Equi Decoder                                |  99   | 86     |    38  |   65    |
> > | RAVEN w/o Equi Encoder & Equi Decoder     |  90   | 56     |    27  |   60    |

---

> > > ### Author Response · Authors · 2025-11-30
> > >
> > > [Part 3 of 3]
> > > > Failure analysis: It would be useful to include qualitative failure cases — for instance, where equivariance assumptions break or where viewpoint perturbations cause prediction drift — to better understand the model’s limits.
> > >
> > > We added a short description of common failure modes observed in the real-world experiments to the updated paper (Appendix H). Most failure modes are due to inaccurate motions (failing to grasp the screwdriver in BoxCleanup) or incorrect subtask sequencing (getting stuck repeating the same behavior like pouring beans in BeansScooping). We did not observe any failures that could be directly attributable to the equivariance assumptions. One side-effect of our method’s SE(3) equivariance is that the gravity direction cannot be distinguished; so one could imagine a task like balancing a weirdly shaped object that could cause issues (we could fix this by adding the gravity vector as an observation).
> > >
> > > With regards to viewpoint perturbations, our experiment in Section 5.1 shows how varying the pose of the agentview camera has a noticeable impact on performance. We have also added another experiment that evaluates the effect of camera calibration errors on performance. Since RAVEN uses the camera parameters to map images to ray features, the calibration accuracy will affect how well it can construct a 3D understanding of the scene. We found that RAVEN is robust to reasonable errors in camera intrinsics and extrinsics, especially when data augmentation of these parameters is done in training. In this setting, RAVEN’s success rate drops only 1% averaged across four MimicGen tasks, when the camera calibration is inaccurate. See the Appendix of the revised paper for full details.
> > >
> > > [1] Chi, Cheng, et al. "Universal manipulation interface: In-the-wild robot teaching without in-the-wild robots." arXiv preprint arXiv:2402.10329 (2024).
> > >
> > > [2] Zhao, Tony Z., et al. "Learning fine-grained bimanual manipulation with low-cost hardware." arXiv preprint arXiv:2304.13705 (2023).
> > >
> > > [3] O’Neill, Abby, et al. "Open x-embodiment: Robotic learning datasets and rt-x models: Open x-embodiment collaboration 0." 2024 IEEE International Conference on Robotics and Automation (ICRA). IEEE, 2024.
> > >
> > > [4] Khazatsky, Alexander, et al. "Droid: A large-scale in-the-wild robot manipulation dataset." arXiv preprint arXiv:2403.12945 (2024).
> > >
> > > [5] Miyato, Takeru, et al. "Gta: A geometry-aware attention mechanism for multi-view transformers." arXiv preprint arXiv:2310.10375 (2023).

---

### Author Response · Authors · 2025-12-03
**Main Response**

We appreciate the reviewers’ time and constructive comments, and thank the emergency AC for working under the time constraints to review our work.  Our paper introduces RAVEN, a novel robot learning method which achieves end-to-end SE(3) equivariance from RGB images by representing them as features attached to rays.  Multiple reviewers highlighted the importance and novelty of this contribution.  To make it easier to get caught up on the rebuttal discussion, we provide a concise summary of the reviews and responses below.

### Reviewer rWp8
**Praise**: Recognizes our method’s significance (“bridging a long-standing gap”) and strong empirical results (“outperforms strong baselines across multiple tasks”).\
**Concerns**: Dependence on known camera parameters; missing ablations.\
**Response**: We added an experiment showing that calibration errors reduce performance by only  1% when trained with data augmentation.  We also added an ablation study demonstrating the importance of RAVEN’s geometric structure and equivariant layers.

### Reviewer DW7R
**Praise**: Highlights novelty (“first ray-based equivariant robot policy”) and scalability (“compatible with more commonly used operations that are heavily optimized”).\
**Concerns**: Equivariance is weaker than other works; unclear empirical benefits of equivariance.\
**Response**: We clarified the equivariance definition and distinguished our work from 3D-input equivariant methods.  New ablations show that encoder and decoder equivariance contribute 18% and 4% to success rate, respectively.

### Reviewer q8H3
**Praise**: Commends the method as “creative solution to a real problem in robotics” and appreciates experiment breadth (“experiments cover a range of test settings”).\
**Concerns**: Equivariance analysis is incomplete; perceived contradiction between equivariance and results.\
**Response**: We pushed back on the concern, clarified the reviewer’s misunderstanding, and improved clarity in the text.  We expanded the equivariance analysis to discuss independent camera movements and added an empirical study on ray-patch discretization.

### Reviewer KKzi
**Praise**: Describes our method as “clear and principled route to achieving SE(3)-equivariance”.\
**Concerns**: Modest improvements over equivariant baselines. \
**Response**: We added equivariant baseline results to DexMimicGen, where our method outperforms by 18% on average, and we show parity with a voxel-based equivariant baseline that uses multiple depth sensors.

After revisions, the paper now includes:
- a more complete and rigorous equivariance analysis,
- new experiments addressing calibration robustness,
- additional ablations isolating where equivariance matters, and
- expanded baseline comparison confirming RAVEN’s empirical advantages.

Given the strengthened analysis and the positive consensus on novelty and significance, we believe the paper now fully meets the ICLR acceptance criteria. We respectfully submit that all substantive concerns have been resolved.

---

### Meta-Review · Area_Chair_biiG · 2026-01-07

**Summary:**

Reviewers broadly acknowledge the novelty and technical soundness of RAVEN’s ray-based approach to achieving SE(3)-equivariance from RGB inputs, with strong empirical results in simulation and limited real-world settings.

However, key concerns converge on 1) strong reliance on perfect camera calibration, 2) lack of ablations and baseline comparisons, 3) ambiguity in the definition and scope of equivariance, and 4) limited real-world experiments.

**Reviewer Concerns:**

- Camera calibration issue is clarified and validated with new experiments.
- Ablations are added, including on ray-based components, the equivariant encoder/decoder, policy heads, and RGB-D inputs; ISP and additional EquiDiffPo results are provided.
- The definition of equivariance is clarified; Ablations isolating the effect of equivariance are addressed with new experiments.
- Failure analysis is included; However, real-world experiments are not expanded.

**Reviewer Scores:**

rWp8 is likely to increase the score from 4 to 6. The requested new results under imperfect extrinsics, with RGB-D, and without equivariance are provided. Failure modes are discussed.


DW7R may increase or maintain the score. Clarificationregarding the scope of SE(3) equivariance and the motivation of ray-based representation are discussed. However, the authors are still encouraged to use a more precise and qualified phrasing for the claim of being "the first SE(3)-equivariant ..." method.


q8H3 is likely to increase the score. Most concerns are addressed, including discussions of equivariance and the ray-based formulation, new results on inference cost, policy heads, and an additional baseline. Real-world experiments are defended but not expanded, while failure cases are now discussed.


KKzi may increase or maintain the score. All requested experiments are provided. However, the discussion around the degree of novelty remains open.

---

### Decision · Program_Chairs · 2026-01-26

Accept (Poster)